# Progressive Memory Transformer: Memory-Aware Attention for Time-Series

## Abstract

Self-supervised learning has become the de-facto strategy for time-series domains where labeled data are scarce, yet most existing objectives emphasize *either* local continuity *or* global shape, seldom both. We introduce **Progressive Memory Transformer** (PMT), a memory-augmented transformer backbone that maintains a writeable memory bank across overlapping windows, allowing representations to accumulate evidence from short, medium, and long horizons without re-reading the entire sequence. On top of our proposed memory-aware attention, we formulate a hierarchical contrastive protocol that aligns embeddings at three complementary granularities—tokens, windows, and full sequences—through a token-window Gaussian loss, a memory-state loss, and a global [CLS] loss. Together, PMT and these multi-scale objectives yield a task-agnostic model for time-series data, providing strong features even when only 1–5% of labels are available. We validate the approach on seven UCR/UEA/UCI benchmarks on classification tasks.

## 1 Introduction

Time-series in geophysics, wearables, finance, and industrial monitoring exhibit structure at *multiple temporal scales* while supervision is *sparse and noisy*. Recent self-supervised (SSL) approaches narrow this supervision gap with objectives that emphasize different parts of the temporal hierarchy, from local timestamp smoothness to instance-level agreement (Chen et al., 2020; Eldele et al., 2021; Lee et al., 2024; Oord et al., 2018; Yue et al., 2022). E.g., autoregressive methods (Oord et al., 2018) emphasize local forecasting, whereas instance-level schemes (Chen et al., 2020) treat an entire sequence as one instance, potentially overlooking within-series locality. Yet, most pipelines still face a core tension: methods that protect *fine-scale* details often fail to propagate *long-range* context without repeated re-encoding, whereas hierarchical pooling trades away local nuance when aggregating across resolutions.

We contend that an effective time-series backbone should (i) *preserve fine-grained patterns*, (ii) *progressively integrate context* as the receptive field expands, rather than re-reading the full past, and (iii) *expose multiple levels: local, medium and global* for contrastive supervision. While standard transformers are stateless, existing memory-augmented architectures often rely on read-only caches, e.g., Transformer-XL (Dai et al., 2019) or fixed latent bottlenecks (Jaegle et al., 2021) that cannot adaptively summarize evolving temporal dynamics.

To bridge this gap, we propose **Progressive Memory Transformer** (PMT), a *stateful* backbone that equips each overlapping window with a compact, *writable* memory and learns when to *retain, refine, or reset* it. High-level context, thus, accumulates *progressively* across windows and depth while preserving fine-scale token detail, reducing the need to re-encode the full past. We refer readers to Fig. 1 for the dataflow and to Section 2 for the precise mechanics (gating, propagation, and overlap aggregation).

To align the representations exposed by PMT without compressing token shape, we couple three complementary contrastive losses: (i) *Hierarchical Gaussian* contrastive loss (HGCL) that promotes local smoothness among nearby tokens and consistency across overlapping windows, (ii) *memory*

Anonymous code: https://anonymous.4open.science/r/PMT-anon-366C

contrastive loss that directly supervises the writable memory tokens (mid-range motifs), and (iii) a *class token* `[CLS]` loss that enforces sequence-level agreement between augmented views. This triad targets local, mid-range and global abstractions, respectively, matching the architecture's three hierarchical elements.

**Contributions.** (1) We propose PMT, a causally masked, memory-augmented backbone whose *window-aligned writable memory* expands receptive field progressively without re-encoding the full past and with a learnable reset gate to handle regime changes. (2) We formulate a multi-scale contrastive protocol that supervises tokens (*HGCL*), memory states (*PCL*), and sequence summaries (*ICL*) without pooling away fine detail. (3) On seven UCR/UEA/UCI benchmarks in low-label settings (1–5%), PMT yields competitive linear-probe results and qualitative evidence that memory states capture mid-range semantics; we also report targeted ablations of the loss components and analyze failure cases to motivate adaptive tokenization.

**Positioning relative to prior work.** Current SSL methods, for time-series, either trade fine-scale detail for broader context after aggressive pooling (Lee et al., 2024; Yue et al., 2022) or lack an explicit mechanism to connect local and global views (Chen et al., 2020; Oord et al., 2018). Previous stateful transformers reduce repeated recomputation typically do not expose their internal states to the learning signal. PMT couples a small, writable memory with losses that supervise each stage of the representation hierarchy. In particular, PMT's memory tokens are (i) **window-aligned** rather than global (ii) **writeable** and propagated both horizontally (window-to-window) and vertically (layer-to-layer), and (iii) **directly supervised** by a contrastive loss (PCL), turning the cache itself into mid-range representations rather than an opaque optimization device. Related work is detailed in Section 4.

## 2 PROGRESSIVE MEMORY TRANSFORMERS

Our primary objective is to robustly represent time-series, i.e., to extract representations that are consistent through the spatiotemporal and semantic scales of the signal. Waveforms are rich in multiscale information. From amplitude spikes, phase shifts, seasonality, and long term dependencies. Thus, the conservation of *local nuances*, *mid-range motifs* and *global context* is critical to produce robust and generalizable features. Our proposal is to have a flexible architecture that uses attention to systematically aggregate neighborhoods through time and over the semantic hierarchies with the use of a memory that retains and discards the previous states of the sequence. Our proposal imbues a transformer with memory that allows it to retain, maintain or discard the states of the processed signal, thus, effectively aggregating the local neighborhoods to produce locally informed global representations.

In the following, we propose to adapt the attention mechanism to be memory-aware that not only aggregates representations spatiotemporally but also stacks them within a semantic hierarchy. We call this variation of attention a Progressive Memory Attention (PMA) block. Moreover, we introduce a set of losses that regularize the set of memory-aware PMAs at dif-

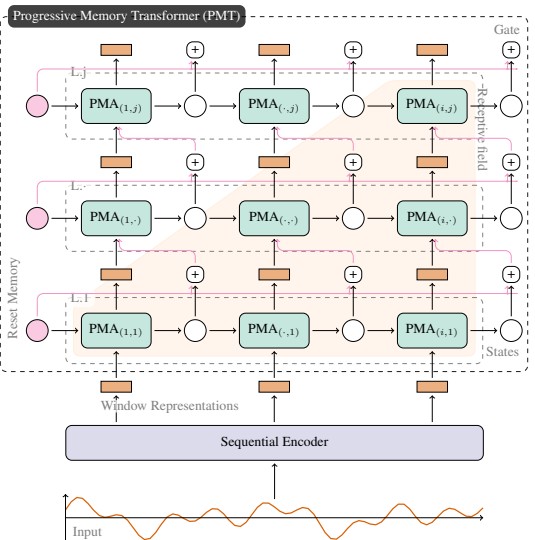

**Figure 1:** An illustration of PMT unrolling several PMA blocks through time (left to right) and hierarchy levels (bottom to top). The sequence is tokenized and encoded through an encoder. Then, each PMA block uses these representations and previous states to produce a new version of both of these inputs. At each level, a given PMA block uses a gating mechanism to control how much of the previous level's memory is passed forward or reset (i.e., uses a reset memory state instead).

ferent granularities. Finally, we summarize our overall pipeline comprising the set of PMA blocks, gating mechanisms and losses that collectively we termed Progressive Memory Transformer (PMT).

**Notation.** Let $x \in \mathbb{R}^{T \times C}$ denote a single $C$-channel time-series of length $T$. The encoder $f_\theta$ produces a matrix $R \in \mathbb{R}^{K \times D}$ containing $K$ *tokens*, each $D$-dimensional, plus a separate `[CLS]`

vector $c \in \mathbb{R}^D$. Hence, the overall output is $[R; c] \in \mathbb{R}^{(K+1)\times D}$. We use the shorthand $f_\theta(\cdot; i)$ to refer to the $i$th output token processed by the encoder.

## 2.1 Progressive Memory Attention

Existing sliding-window transformers (Nie et al., 2023) either discard earlier windows or re-encode them every step, whereas Transformer-XL (Dai et al., 2019) caches the past in a *read-only* form. In contrast, we propose the **Progressive Memory Attention (PMA)** that equips each window with a *writable*, fixed-size memory. Unlike read-only caches, writability enables *adaptive summarization*: the model learns to promote, refine, or overwrite context as new evidence arrives, and we supervise these states directly via a window-level contrastive loss (Section 2.2.1). The memory propagates *horizontally* to the next window and *vertically* up the stack, so the receptive field widens by the stride $S$ as the window advances while deeper layers fuse these compact summaries without re-encoding all tokens, see Fig. 2 and Eq. (1).

For a given window $i$ at a level $j$, we compute a window $W_{i,j}$ and memory $M_{i,j}$ representations through our proposed PMA block, such that

$$W_{i,j}, M_{i,j} = \text{PMA}(M_{i-1,j}, \bar{M}_{i,j-1}, \bar{W}_{i,j-1}), \quad (1)$$

where $M_{i-1,j}$ provides the *temporal* context (memory from the previous window at the same level), and $\bar{M}_{i,j-1}$ and $\bar{W}_{i,j-1}$ provide *hierarchical* context (refined memory and window from the previous level). The initial memory for level $j$ is a *reset state* used for initialization and adaptive reset, i.e., $M_{0,j} = M_r$. We refine the memory at a given window and level with a learnable gate that mixes the carry-over memory and the reset state (implemented as a pre-update mix that forms $\bar{M}_{i,j}$)

$$\bar{M}_{i,j} = G_M(M_{i,j}, M_r), \quad (2)$$

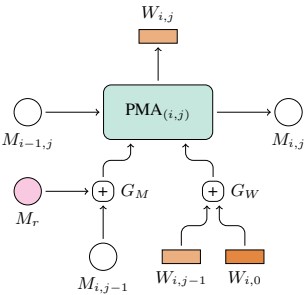

**Figure 2:** An illustration of the information flow (1) in a PMA block.

that adaptively mixes the memory with the reset token $M_r$, enabling the model to overwrite stale context and prevent drift when regimes change. This is reminiscent of forget/reset gates in LSTMs (Hochreiter & Schmidhuber, 1997), but applied to window-level memory tokens updated via attention rather than to per-time-step hidden states. Similarly, we refine the representations with an adaptive gating function,

$$\bar{W}_{i,j} = G_W(W_{i,j}, W_{i,0}), \quad (3)$$

that mixes the original signal representation, $W_{i,0} = f_\theta(x; i)$, with the processed one $W_{i,j}$ to preserve local detail while integrating context. We illustrate this block in Fig. 2. Within each PMA block we apply one multi-head FlashAttention-2 (Dao, 2024) layer to the concatenated input sequence (1). Memory queries are unmasked; window queries see every key in $M_{i-1,j}$ and the strictly-causal slice of their own window but are blocked from $\bar{M}_{i,j-1}$ preventing leakage from deeper (potentially future-aware) summaries and keeping updates strictly autoregressive.

We perform this computation (1) for every window and level in a forward manner. Afterward, we process a reversed copy of the sequence with an independent PMA stack and fuse the two token streams after both passes complete, thereby keeping the forward computation strictly unidirectional.

In our implementation, we normalize the representations layer-wise. Moreover, we mask the sequence at each level to attend to the corresponding neighborhoods using FlashAttention-2; see Section 2.3 for details.

**Attentive overlap aggregation & residual path.** Because windows overlap, the same position appears $O$ times per block; an overlap pooler merges these rows so the token length stays constant across blocks. The pooler is a single-query cross-attention per position over its $O$ overlapped patches (keys/values from the overlaps) with a learned head×overlap bias. Its output is scaled by $1/\sqrt{O}$, post-normalized, and multiplied by a learnable $\gamma$, then fused with the masked-mean skip via a SkipGate before re-entering the residual stream. Standard pre-norm skips wrap the chunk-processing and feed-forward sub-layers. Additional gates control state reuse across blocks and, in the bidirectional model, the fusion of the two directional streams.

**Receptive-field growth.** A formal derivation (Appendix B) shows that after $B$ PMA blocks the contextual span of a token expands linearly with both block depth and token index.

## 2.2 MULTI-TASK LOSSES

Different downstream signals stand out at different temporal scales, e.g., a premature heartbeat is a half-second glitch, whereas a walking about is a multi-second pattern. We, therefore, attach three contrastive losses, each nudging the representation at a specific granularity. One (hierarchical loss) smooths neighboring patch tokens and overlaps, so fine-scale details remain coherent even when phase-shifted. The second one (memory loss) aligns the few writable tokens that summarize each window, preventing collapse and preserving mid-range context that survives our augmentations. Finally, the last one (class loss), applied after neighborhood masking, forces every local token to contribute whatever information is globally predictive, yielding a stable sequence-level descriptor. Each interface of the architecture exposes a different temporal scale (patch tokens, memory tokens, [CLS]); supervising all three, without compressing token shape, aligns local nuance, mid-range motifs, and global semantics in one backbone.

**Two-view augmentation.** Following established protocols in time-series contrastive learning (Eldele et al., 2023; 2024; Yue et al., 2022), we present *two* stochastic views of every sample to a weight-shared backbone: a **weak** view that preserves the signal's local morphology (channel-wise z scaling and low-variance Gaussian noise) and a **strong** view that, *additionally*, breaks short-range correlations while keeping coarse semantics (time-warping and magnitude-warping).

**Hierarchical targets.** Our model treats a time-series as a hierarchy of progressively coarser views: *tokens windows*, and the full sequence (global context). Within each PMA block, the writable memory bank decides, via attention, which local details to persist, refine, or overwrite, so deeper blocks can re-use distilled high-level cues while still observing fresh low-level evidence. We, therefore, attach learning signals at three complementary abstraction levels: *token-wise*, *window-wise* and *sequence-wise*. This multi-granular supervision forces the the backbone to align representations that are simultaneously sensitive to fine temporal nuances and consistent across wider temporal spans, providing rich anchors for downstream heads. Here, we will describe how each loss component (**HGCL**) for token/window alignment, and two contrastive objectives on the PMA memory tokens (**PCL**) and the [CLS] token (**ICL**) operationalize this hierarchy.

### 2.2.1 PMA CONTRASTIVE LOSS

The PMA Contrastive Loss (PCL) acts on the set of *memory-state tokens* produced by the PMA backbone and is applied at the **window level** of the hierarchy. Each memory token is first passed through a view-shared, two-layer projection head $g(\cdot)$ and then $\ell_2$-normalized. Supervising the memory tokens directly trains the model *what to remember*. Salient mid-range evidence must be distilled into the writable slots consistently across views.

Let $B$ be the batch size, $v \in \{1, 2\}$ the view index, $w = 1, \ldots, W$ the window index, and $\ell = 1, \ldots, H$ the PMA-level index. Let the $b$th sample's memory representation for the view $v$ at position $(w, \ell)$ be $M_{b,w,\ell}^{(v)}$, then we define the set $M^{(v)} = \left\{ M_{b,w,\ell}^{(v)} \right\}_{b=1,\ldots,B;w=1,\ldots,W;\ell=1,\ldots,H}$ with each matrix $M_{b,w,\ell}^{(v)} \in \mathbb{R}^{n_m \times D}$ holding the $n_m$ memory tokens of that window and level. For an *anchor* token $m_a = M_{b,w,\ell}^{(1)}[k]$, $k = 1, \ldots, n_m$, the positive is $m_p = M_{b,w,\ell}^{(2)}[k]$, while negatives come from $N_a = \left\{ m_n \in M^{(2)} : m_n \notin b \right\}$, i.e., all tokens of the opposite view that originate from a *different* sequence in the mini-batch. The resulting InfoNCE (Oord et al., 2018) loss is

$$\mathcal{L}_{\text{PCL}} = -\frac{1}{BWHn_m} \sum_{m_a \in M^{(1)}} \log \frac{\exp\big(\langle g(m_a), g(m_p)\rangle/\tau\big)}{\exp\big(\langle g(m_a), g(m_p)\rangle/\tau\big) + \sum_{m_n \in N_a} \exp\big(\langle g(m_a), g(m_n)\rangle/\tau\big)}. \quad (4)$$

All vectors are unit-normalized, so $\langle \cdot, \cdot \rangle$ is the cosine similarity; and $\tau$ is a fixed temperature. Because each window holds only $n_m \ll K$ memory tokens (where $K$ is the patch-token count), the loss can be evaluated over *all* tokens without subsampling. We compute this loss for the forward and backward passes and average them, thus, we use the loss $\mathcal{L}_{\text{PCL}}^* = \frac{1}{2}\big(\mathcal{L}_{\text{PCL}}^{\text{fwd}} + \mathcal{L}_{\text{PCL}}^{\text{bwd}}\big)$.

### 2.2.2 INSTANCE CONTRASTIVE LOSS

The token stream from the PMA backbone is refined by $L$ Transformer layers with *neighborhood masking*: token at position $t$ attends only to positions $t - n, \ldots, t$, whereas the [CLS] token itself

attends to every position. This constraint maintains quadratic capacity for aggregation tokens while bounding the cost for patch interactions to $\mathcal{O}(L\,nT)$. The encoder, thus, (i) sharpens local patch features through short-range context and (ii) aggregates global summaries into the `[CLS]` token, supporting both instance contrastive-learning and downstream classification.

For a mini-batch of $B$ time-series we generate two stochastic views, yielding $2B$ sequences and therefore $2B$ `[CLS]` embeddings. Denote the `[CLS]` token produced by view $v \in \{1, 2\}$ of sequence $b \in \{1, \dots, B\}$ by $\mathbf{c}_b^{(v)} \in \mathbb{R}^D$. Each is passed through a *`[CLS]`-token-specific two-layer projection head* $g(\cdot)$ (shared across views) and $\ell_2$-normalized. Thus, the loss is

$$\mathcal{L}_{\text{ICL}} = -\frac{1}{B} \sum_{b=1}^{B} \log \frac{\exp\big(\langle g(\mathbf{c}_b^{(1)}), g(\mathbf{c}_b^{(2)}) \rangle / \tau\big)}{\sum_{\substack{b'=1 \\ (b',v) \neq (b,1)}}^{B} \sum_{v=1}^{2} \exp\big(\langle g(\mathbf{c}_b^{(1)}), g(\mathbf{c}_{b'}^{(v)}) \rangle / \tau\big)} \tag{5}$$

Thus each anchor $\mathbf{c}_b^{(1)}$ pairs with its positive $\mathbf{c}_b^{(2)}$, while the remaining $2(B-1)$ `[CLS]` tokens in the batch act as negatives. This instance-based contrastive-loss constitutes the *sequence-wise* component of our loss protocol. We explored a queue-based memory bank but found no performance improvements; hence, all experiments use only in-batch negatives.

### 2.2.3 HIERARCHICAL GAUSSIAN CONTRASTIVE LOSS

Unlike TS2Vec (Yue et al., 2022), TNC (Tonekaboni et al., 2021), and the temporal arm of SoftCLT (Lee et al., 2024), which confine positives to within-series neighborhoods (or their soft surrogates) and typically control cost by pooling or striding, our HGCL preserves full token shape and creates richly diverse pairs via Gaussian-weighted token/window positives with instance-wise in-batch negatives. HGCL keeps full token shape and uses instance-wise negatives (in-batch across sequences) while defining *soft* positives at two scales: (i) *token-level* smoothness within a window weighted by a Gaussian over temporal distance, and (ii) *window-level* consistency for overlapping windows weighted by a Gaussian over stride multiples—Eqs. (6) and (7).

At the *token-level*, embeddings within the same window are treated as *soft positives*, with similarity weighted by temporal proximity according to a Gaussian function. At the *window-level*, embeddings representing overlapping windows are also considered soft positives, promoting temporal consistency at broader scales due to overlap in the sliding window approach.

Formally, the Gaussian weighting schemes at each level are defined as follows:

$$w_{ij}^{\text{patch}} = \exp\left(-\frac{(i-j)^2}{2\sigma_p^2}\right), \quad w_{uv}^{\text{window}} = \exp\left(-\frac{\big((u-v) \cdot S\big)^2}{2\sigma_w^2}\right), \tag{6}$$

where indices $i, j$ represent positions within a local window at the patch-level, and $u, v$ denote window indices separated by stride $S$ at the window-level. Parameters $\sigma_p$ and $\sigma_w$ control the Gaussian weighting sharpness at the respective scales.

The HGCL objective combines these two scales into a unified InfoNCE loss:

$$\mathcal{L}_{\text{HGCL}} = \alpha \mathcal{L}_{\text{patch}} + \beta \mathcal{L}_{\text{window}}, \tag{7}$$

where $\alpha$ and $\beta$ control the balance between fine-grained local similarity and coarser temporal consistency.

Negative samples for contrastive training are uniformly drawn from other sequences within the training batch (in-batch negatives), ensuring robust discriminative learning. Through this hierarchical Gaussian-weighted approach, HGCL promotes learning of temporally coherent representations across multiple resolutions, bridging local alignment and broader contextual consistency.

### 2.3 PIPELINE SUMMARY

**Pipeline in five steps.** (1) A 1-D convolution tokenizes a $C$-channel waveform into $K$ patch tokens (plus a `[CLS]` vector); we keep tokens uncompressed thereafter. (2) We unfold tokens into windows of length $W$ and stride $S$ and run $B$ PMA blocks; at each block the writable memory flows horizontally (window-to-window) and vertically (level-to-level). (3) An attentive overlap-pooler

merges duplicates so the token length remains constant across blocks. (4) We append `[CLS]` and apply a short stack of *locally masked* encoder layers (`[CLS]` attends globally). (5) We train with three **complementary** objectives: **HGCL** on tokens/windows, **PCL** on memory tokens, and **ICL** on sequence summaries. Implementation details (window normalization across datasets, masking, and vectorized kernels) appear in Appendices C and D.

## 3 EXPERIMENTS

### 3.1 DATASETS AND EVALUATION PROTOCOL

We evaluate PMT on seven time-series classification benchmarks—UCR (Dau et al., 2018), UEA (Bagnall et al., 2018), and UCI (Anguita et al., 2013): *HAR*, *Epilepsy*, *Wafer*, *FordA*, *FordB*, *Phalanges-OutlinesCorrect (POC)*, and *ElectricDevices*. These datasets were chosen to jointly satisfy two criteria: *diversity* (domains, lengths, channel counts) and *sufficient unlabeled volume* for contrastive objectives that rely on in-batch negatives. Across the seven datasets, the training splits comprise $\sim$21.5M time-steps (i.e., $> 40\%$ of the combined UCR+UEA volume), while covering a broad range of sequence lengths and class cardinalities (Appendix A). In InfoNCE-style training, representation quality improves with the number of in-batch negatives (Chen et al., 2020; Oord et al., 2018). We, therefore, avoid external memory banks or queues (He et al., 2020; Wu et al., 2018) to keep a single, comparable training recipe across datasets; as a result, many micro-datasets in UCR/UEA (e.g., *GunPoint*, *CBF*) with only tens of sequences are unsuitable for our large-batch protocol.

We follow a standard linear-evaluation protocol (Eldele et al., 2021): the backbone is pretrained without labels on each training set; we then train an SVM probe (matching Lee et al., 2024) on $1\%$ and $5\%$ labeled subsets while freezing the backbone, and report top-1 accuracy and macro-F1 on the official test split.

We compare against time-series SSL baselines TS2Vec (Yue et al., 2022), TS-TCC (Eldele et al., 2021), and SoftCLT (Lee et al., 2024), as well as generic SSL baselines adapted to time-series: CPC (Oord et al., 2018), SimCLR (Chen et al., 2020), and SSL-ECG (Sarkar & Etemad, 2020). Related work appears in Section 4.

### 3.2 MAIN RESULTS ON SELF-SUPERVISED CLASSIFICATION

Table 1 presents the classification performance of our method versus the baselines on each dataset, for $1\%$ and $5\%$ labeled training data. Our approach achieves competitive results across all datasets. The most noteworthy result is the performance on the HAR dataset, where our method's $1\%$ labels we surpass the $5\%$-label performance of prior methods on HAR. In Section 3.4, we explore the performance on this dataset deeper.

**Table 1:** Self-supervised learning results on $1\%$ and $5\%$ labeled subsets. Each cell shows accuracy / macro-F1.

| Dataset | SSL-ECG | CPC | SimCLR | TS2Vec+SoftCLT | TS-TCC+SoftCLT | PMT |
|---|---|---|---|---|---|---|
| **Self-supervised learning (1% labeled)** | | | | | | |
| HAR | 60.0 / 54.0 | 65.4 / 63.8 | 65.8 / 64.3 | 91.0 / 91.0 | 82.9 / 82.8 | **92.9 / 93.2** |
| Epilepsy | 89.3 / 86.0 | 88.9 / 85.8 | 88.3 / 84.0 | 96.3 / 94.1 | 95.6 / **95.6** | **96.5** / 94.4 |
| Wafer | 93.4 / 76.1 | 93.5 / 78.4 | 93.8 / 78.5 | 95.3 / 88.1 | 96.5 / 96.5 | **98.9 / 97.1** |
| FordA | 67.9 / 66.2 | 75.8 / 75.2 | 55.9 / 55.7 | 87.1 / 87.1 | 81.5 / 81.2 | **88.2 / 88.2** |
| FordB | 64.4 / 60.5 | 66.8 / 65.0 | 50.9 / 49.8 | 67.9 / 67.9 | 74.8 / 74.8 | **76.3 / 76.2** |
| POC | 62.5 / 41.2 | 64.8 / 48.2 | 61.5 / 38.4 | 63.6 / 62.8 | 65.4 / 64.6 | **68.4** / 40.6 |
| ElectricDevices | 60.1 / 50.0 | 59.3 / 48.9 | 62.5 / 51.2 | 62.0 / 53.0 | **64.6 / 63.2** | 59.0 / 52.2 |
| **Self-supervised learning (5% labeled)** | | | | | | |
| Dataset | SSL-ECG | CPC | SimCLR | TS2Vec+SoftCLT | TS-TCC+SoftCLT | PMT |
| HAR | 63.7 / 58.6 | 75.4 / 74.7 | 75.8 / 74.9 | 92.1 / 92.1 | 92.6 / 92.6 | **95.5 / 95.9** |
| Epilepsy | 92.8 / 89.0 | 92.8 / 90.2 | 91.3 / 89.2 | 96.7 / 94.9 | 96.2 / **96.1** | **96.7** / 94.9 |
| Wafer | 94.9 / 84.5 | 92.5 / 79.4 | 94.8 / 83.3 | 98.8 / 96.8 | 98.2 / **98.2** | **99.2** / 97.8 |
| FordA | 73.6 / 70.7 | 86.5 / 86.5 | 69.6 / 68.9 | 92.5 / 92.5 | **93.2 / 93.2** | 89.5 / 89.5 |
| FordB | 71.7 / 69.8 | 86.3 / 86.2 | 63.0 / 60.7 | 78.8 / 78.6 | **88.0 / 88.0** | 76.7 / 76.7 |
| POC | 62.9 / 43.3 | 66.9 / 44.3 | 62.7 / 42.4 | 70.9 / 69.7 | 69.4 / 66.3 | **74.5 / 71.8** |
| ElectricDevices | 63.7 / 56.1 | 62.4 / 58.1 | 63.9 / 58.6 | 62.4 / 54.4 | 65.1 / **63.8** | **65.1** / 58.7 |

tion 3.4, we explore the performance on this dataset deeper. The table also highlights the engine-vibration benchmarks FordA and FordB, whose labels depend on short-lived phase and frequency shifts over only a few time steps. These datasets are particularly sensitive to how the waveform is tokenized. Section 3.4.1 analyzes this effect and shows that performance is governed mainly by the stride (overlap) of the patchified input rather than the patch length itself. With a tokenizer that uses moderately sized patches and small stride, PMT remains competitive on FordA and FordB in the $1\%$-label setting, but still lags the strongest baselines in the $5\%$ regime. Overall, PMT outperforms the other baselines in average Top-1 accuracy in the $1\%$-label scenario (82.9% vs. 80.5% for TS2Vec+SoftCLT). For completeness, we ran this experiment replacing the PMA blocks with xLSTM and with Transformer-XL in the appendix (H)

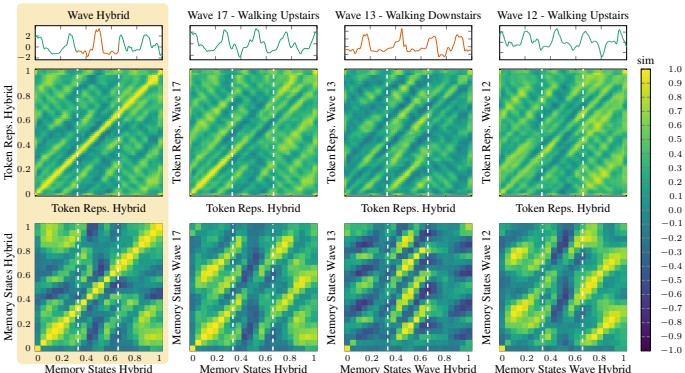

**Figure 4:** Cosine similarity matrices for the representations and states between pair-wise signals from HAR. Higher similarity shows that the signals correlate as evidenced by the learned embeddings. The vertical bars denote the different sections of the hybrid wave. The wave number corresponds to the sample index in the test dataset.

### 3.3 PMA MEMORY VISUALIZATION

We probe the embeddings of the proposed encoder at two resolutions: *local nuance* (token representations) and *medium range* (PMA memory states). We center the qualitative discussion on the human-activity-recognition (HAR) data because its multichannel inertial traces exhibit macroscopic structures, most notably the periodic peaks of foot-strikes, that are visually intelligible even without signal-processing expertise. Such "readable" waveforms allow us to relate patterns in the similarity maps directly to real-world primitives (steps, turns, pauses), making HAR an ideal test-bed for embedding inspection. For completeness, we do a short analysis on embeddings from the epilepsy dataset as well.

Figures 3 and 4, and more in Appendix J, show the token representations, and the memory states over waveforms from HAR spliced from 3 unique waveforms. Figure 3 contains the classes: standing (first and last third) and laying (for data generation details see Appendix C, and for more results see Appendix J). We plot only one channel of the 9 here for interpretability, while the model processed all. As the PMA unrolls over windows of tokens, with a stride greater than one, we have more token representations (*local nuance*), than memory states (*mid-range motifs*).

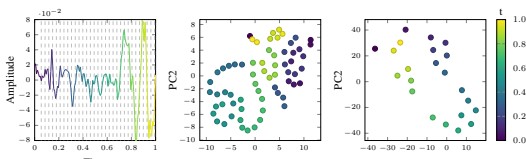

**Figure 3:** (Left) Double spliced HAR waveform. (Middle) TSNE of representations (tokens) of the signal. (Right) TSNE of memory-states. The splicing is sourced from two unique waveforms from separate activities (standing and laying)

The center subplot of Fig. 3 highlights the ability of PMT to extract token representations that are semantically consistent with the underlying signal. This is observed by the linear separability of the tokens based on their source. This observation extends to the memory-state figure. We note that the first memory-state in the sequence is often less semantically significant due to its small effective receptive field, see Appendix B, and its proximity to the initial reset state. We have omitted this memory state token from the figures.

Figure 4 show the cosine similarity matrix of the token representations and the memory states (from the last PMA block) representations from three unique waveforms spliced together and processed by PMT. The two unique classes of the waveform were chosen for their semantic proximity: walking upstairs, walking downstairs, and walking upstairs. We plot a single channel, although the model sees all. The top cosine similarity matrix show the token representation similarity matrix, while the bottom show the memory state tokens. The leftmost figure show the self similarity matrix using the hybrid/spliced waveform. The figures to the right show token and memory state embedding comparisons from the hybrid waveform and the original.

In the left (orange highlight) self-similarity matrix the main diagonal is expected, but the block-diagonal "checkerboard" pattern reveals that embeddings inside the same activity segment cluster tightly, while inter-segment pairs are pushed apart. The second and third blocks—both *walking*

**Figure 5:** Cosine similarity matrices of the representations and states for the Epilepsy dataset. Comparison of two sequences from the same class 1 (left) and class 2 (middle), and between two sequences from different classes (right).

**Table 2:** Impact of $\lambda_{(\cdot)}$ on HAR and FordA, averaged over $1\%$ and $5\%$ label splits. Higher is better.

| (a) $\lambda_{HGCL}$ ablation HAR. | | | (b) $\lambda_{HGCL}$ ablation FordA. | | | (c) $\lambda_{PCL}$ ablation HAR. | | | (d) $\lambda_{PCL}$ ablation FordA. | | |
|---|---|---|---|---|---|---|---|---|---|---|---|
| $\lambda_{HGCL}$ | Top-1 ↑ | mF1 ↑ | $\lambda_{HGCL}$ | Top-1 ↑ | mF1 ↑ | $\lambda_{PLC}$ | Top-1 ↑ | mF1 ↑ | $\lambda_{PLC}$ | Top-1 ↑ | mF1 ↑ |
| 0 | 93.35 | 93.68 | 0 | 78.42 | 78.38 | 0 | 93.32 | 93.70 | 0 | **83.91** | **83.91** |
| 0.01 | **93.47** | **93.86** | 0.01 | 75.58 | 75.44 | 0.25 | 93.54 | 93.95 | 0.25 | 83.13 | 83.08 |
| 0.1 | 93.36 | 93.70 | 0.1 | 80.63 | 79.84 | 0.5 | **93.73** | **94.13** | 0.5 | 83.24 | 83.23 |
| 0.5 | 92.84 | 93.16 | 0.5 | 82.90 | 82.90 | 1 | 93.56 | 93.88 | 1 | 82.64 | 82.61 |
| 1 | 92.95 | 93.22 | 1 | **84.32** | **84.31** | | | | | | |

*upstairs*—are mutually bright, indicating class-level invariance to absolute position for both the *local* tokens and *medium-range* memory state tokens. The persistence of this pattern to memory states show that the progressive memory extract class relevant content. When we correlate the hybrid sequence with the original waveforms, only the semantically matching blocks light up (e.g., hybrid-vs-upstairs is bright in blocks 2 and 4 but not in block 3). This indicates that PMT learns representations that are simultaneously *class-separable* and *temporally compositional*.

Figure 5 show 3 sets of cosine similarity matrices from the epilepsy dataset. Each pair consists of a token representation comparison on the left and a memory state comparison on the right. The first column of each pair compares *token embeddings*; the right column compares the corresponding *progressive-memory states*. **Same-class pairs (columns 1–4):** Diagonal stripes are faint at token level but become sharply defined after memory integration, indicating that the progressive-memory layer consolidates class-specific cues while damping phase noise. **Cross-class pair (columns 5–6):** Residual token-level correlations disappear almost entirely in the memory states, suggesting that sequences from different classes are mapped to near-orthogonal regions of latent space. Overall, class separability is mainly realized *after* recurrent aggregation; token embeddings alone retain limited spectral overlap.

## 3.4 ABLATIONS

The ablations in Table 2 show the impact of $\lambda_{(\cdot)}$ on the total loss function: $\mathcal{L}_{Total} = \lambda_{ICL}\mathcal{L}_{ICL} + \lambda_{PCL}\mathcal{L}_{PCL} + \lambda_{HGCL}\mathcal{L}_{HGCL}$. We focus on HAR, as this dataset presents a diverse set of classes with varying degrees of difficulty in terms of class discrimination. To

**Table 3:** Ablation results on HAR and FordA, averaged over $1\%$ and $5\%$ label splits. Higher is better.

| (a) Loss-component ablation HAR. | | | | | |
|---|---|---|---|---|---|
| Setup | $\lambda_{ICL}$ | $\lambda_{HGCL}$ | $\lambda_{PCL}$ | Top-1 ↑ | mF1 ↑ |
| All $\lambda = 1$ | 1 | 1 | 1 | 93.32 | 93.72 |
| Balanced (all) | 1 | 0.1 | 0.25 | 93.54 | 93.95 |
| *Leave-one-out* | | | | | |
| $\lambda_{ICL} = 0$ | 0 | 0.1 | 0.25 | 90.74 | 91.03 |
| $\lambda_{HGCL} = 0$ | 1 | 0 | 0.25 | 92.81 | 93.18 |
| $\lambda_{PLC} = 0$ | 1 | 0.1 | 0 | 86.61 | 86.61 |
| *Single-only* | | | | | |
| ICL only | 1 | 0 | 0 | 92.80 | 93.20 |
| HGCL only | 0 | 1 | 0 | 85.76 | 85.47 |
| PCL only | 0 | 0 | 1 | 91.38 | 91.69 |

| (b) Loss-component ablation FordA. | | | | | |
|---|---|---|---|---|---|
| Setup | $\lambda_{ICL}$ | $\lambda_{HGCL}$ | $\lambda_{PLC}$ | Top-1 ↑ | mF1 ↑ |
| All $\lambda = 1$ | 1 | 1 | 1 | 77.66 | 77.48 |
| Balanced (all) | 1 | 0.25 | 0.5 | 83.24 | 83.23 |
| *Leave-one-out* | | | | | |
| $\lambda_{ICL} = 0$ | 0 | 0.25 | 0.5 | 82.44 | 82.39 |
| $\lambda_{HGCL} = 0$ | 1 | 0 | 0.5 | 76.44 | 76.06 |
| $\lambda_{PLC} = 0$ | 1 | 0.25 | 0 | 83.91 | 83.89 |
| *Single-only* | | | | | |
| ICL only | 1 | 0 | 0 | 71.09 | 70.53 |
| HGCL only | 0 | 1 | 0 | 73.31 | 72.76 |
| PLC only | 0 | 0 | 1 | 72.89 | 71.00 |

contrast the rich multi-scale feature space of HAR, we explore FordA, to highlight a dataset with highly local differentiating features. The Ford datasets present a particularly difficult set of features that our model is less sensitive to. The contrasting ablations yield insight into how emphasizing or deemphasizing the losses that are most critical for success on these datasets. Our ablations consists of training sessions of 500 epochs over the HAR and FordA datasets, with the average of 3 unique seeds for each category. We report average top-1 accuracy and macro-F1 score for $1\%$ and $5\%$ label splits.

The results outlined in Table 3 show that HAR's strong performance is largely dependent on higher level features, produced by the PCL and ICL losses. While a small $\lambda_{HGCL}$ does contribute positively,

this is too small to be significant. FordA's results indicate that a stronger emphasis on the tokens yield a higher return, than the PCL loss, underlining the importance of encouraging local nuance for this dataset. Our three losses behave like complementary *focus knobs*-HGCL sharpens fine-scale texture, PCL condenses mid-range motifs, and ICL aligns global semantics. Adjusting these knobs lets a practitioner steer the encoder towards known signal characteristics in the data.

The less important knobs quickly reveal themselves by their $\Delta$ response. Accuracy remains within 1.27% for $\lambda_{PCL}$, and 8.74% for $\lambda_{HGCL}$, revealing HGCL, and thus *local nuance* to be the knob to pay attention to for FordA. Similarly for HAR, accuracy remains within 0.63 % accuracy for $\lambda_{HGCL}$, and 0.41% for $\lambda_{PCL}$, indicating that $\lambda_{ICL}$ is the most impactful for performance. We have additional ablations in the Appendix F.

### 3.4.1 PATCH AND STRIDE

FordA and FordB are the most sensitive benchmarks to our patchified input stem, because their labels depend on short-lived vibration events. ElectricDevices, while not a vibration dataset, is another benchmark where PMT underper-

**Table 4:** Ablation over patch sizes and stride FordA/B, and ElectricDevices. Training 150 epochs, using the mean 1% and 5% results over 3 seeds.

(a) Keeping a constant stride (9), ablating *patch sizes* as a fraction of the seq. length.

| Patch size | FordA Top-1 | MF1 | FordB Top-1 | MF1 | ElectricDevices Top-1 | MF1 |
|---|---|---|---|---|---|---|
| 2.5% | 82.2 | 79.0 | 70.7 | 66.5 | 60.3 | 53.9 |
| 5% | 85.0 | 83.9 | 72.9 | 70.4 | 60.7 | 53.9 |
| 7.5% | 86.4 | 85.4 | 75.5 | 75.0 | 61.7 | 55.9 |
| 10% | 86.7 | 85.6 | 75.3 | 74.7 | 61.8 | 55.1 |
| 12.5% | 87.4 | 86.7 | 75.4 | 75.1 | 61.5 | 54.4 |
| 15% | 87.4 | 86.0 | 74.8 | 74.1 | 61.1 | 53.8 |

(b) With a constant patch size (62), ablating *stride* as a fraction of the patch size.

| Stride | FordA Top-1 | MF1 | FordB Top-1 | MF1 | ElectricDevices Top-1 | MF1 |
|---|---|---|---|---|---|---|
| 10% | 87.7 | 87.3 | 75.3 | 74.1 | 61.5 | 54.4 |
| 25% | 85.0 | 83.1 | 74.8 | 74.1 | 60.2 | 53.0 |
| 50% | 72.9 | 65.1 | 67.2 | 64.2 | 59.4 | 52.7 |
| 75% | 66.4 | 56.0 | 67.6 | 62.5 | 58.4 | 52.3 |
| 100% | 62.5 | 57.8 | 59.5 | 55.4 | 57.9 | 51.8 |

forms strong baselines in Table 1, so we include it in the same sweep to probe whether tokenizer choices are also a limiting factor. To disentangle tokenizer effects from the PMA blocks, we run a small grid over patch length and stride on these two datasets (Table 4(a)). We keep the backbone and loss weights fixed, train each configuration for 150 epochs, and report the mean over the 1% and 5% label splits and three seeds.

Table 4(b) shows that, with a fixed stride, varying the patch length from very short to longer patches changes accuracy only moderately on all three datasets, indicating a fairly wide plateau once patches are long enough to cover the relevant pattern. In contrast, Table 4(b) shows that fixing the patch size and increasing the stride from heavily overlapping to non-overlapping patches causes a sharp degradation on FordA and FordB and a smaller but still monotonic drop on ElectricDevices. This supports the interpretation that the main failure mode is coarse temporal downsampling: large strides under-sample short-lived events so they never enter the token sequence, and they also hurt ElectricDevices even though its discriminative structure is less localized. In all FordA/B experiments in Table 1 we therefore adopt the small-stride, moderate-patch configuration suggested by this ablation.

## 4 RELATED WORK

**Self-Supervised Learning for Time-Series.** Early contrastive methods such as CPC (Oord et al., 2018) and SimCLR (Chen et al., 2020) learn instance-level representations. CPC (Oord et al., 2018) predicts future latents along the timeline combined with a probabilistic contrastive loss, whereas SimCLR (Chen et al., 2020) treats the entire sequence as a single view and therefore ignores within-series locality. Later work injects explicit temporal structure. For instance, TS-TCC (Eldele et al., 2021) couples *temporal* prediction (cross-view future inference) with *contextual* instance discrimination. CA-TCC (Eldele et al., 2023) keeps these losses but adds a class-aware term for semi-supervised learning—no cross-attention module is introduced. TNC (Tonekaboni et al., 2021) contrasts points inside vs. outside a fixed Gaussian neighborhood to promote local smoothness, while SoftCLT (Lee et al., 2024) soft assigns positive anchors in a Gaussian neighborhood and *soft* DTW-based weights so phase-shifted segments contribute graded positive signal. TS2Vec (Yue et al., 2022) enforces contrastive agreement at multiple *compressed* temporal resolutions, producing scale-specific embeddings that are later pooled.

In contrast, our framework supervises at three levels *without compressing token shape*: (i) a token/window loss (HGCL) that in-batch estimates Gaussian weights, (ii) a memory-token loss (PCL), and (iii) Instance Contrastive Loss (ICL) for sequence-wise alignment. Thus, we preserve fine-

grained cues while adding hierarchical context, avoiding the information loss that can occur when representations are repeatedly pooled or strided as in TS2Vec.

**Memory-Augmented Transformers.** Vanilla Transformers are *stateless*: after a window is processed the past must be re-read. Transformer-XL (Dai et al., 2019) lengthens context by *caching* the previous segment's hidden states and concatenating them to the next, yielding a longer (but strictly *read-only*) horizontal context. Set Transformer (Lee et al., 2019) and Perceiver (Jaegle et al., 2021) introduce learnable auxiliary tokens that travel only *vertically* through layers within a sample, acting as a fixed-size latent bottleneck that is reset every sequence. Sequence Complementor (Chen et al., 2025) applies a related idea to timeseries forecasting by appending a small number of learnable complementary sequences to patchified Transformer encoders; these hlobal tokens are shared across sequences and discarded after the encoder, enriching local tokens but not storing sequence-specific state across windows. Titans (Behrouz et al., 2024) combines both ideas with *writable* persistent slots that survive across segments, but exposes them as a single global bank detached from any sliding-window alignment. PatchTST (Nie et al., 2023) discards memory altogether, using patch tokens and vanilla self-attention; distant dependencies must therefore be supplied in an increasingly long look-back window.

Our *Progressive Memory Attention (PMA)* targets the remaining gap for time-series data. Every block carries a small memory bank that is *refreshed horizontally* from one *overlapping* window to the next and *forwarded vertically* to deeper blocks. This two-track propagation lets the model accumulate local evidence while hierarchically expanding its receptive field—aligning memory flow with the sliding-window regime common in sensor, seismic, and forecasting workloads. PMA therefore blends Transformer-XL's horizon, the latent-token economy of Set Transformer/Perceiver, and Titans' read-write flexibility, while keeping the additional memory state compact and strictly causal.

## 5 CONCLUSION

We introduced the Progressive Memory Transformer as a solution to address the stateless attention issues prevalent in current models for time-series. Given the unique challenges posed by time-series, such as multiple temporal scales and sparse, noisy annotations, it is crucial to equip models with memory and decision-making capabilities for more effective sequence processing. Our proposed memory-aware attention mechanism (PMA) uses a writable memory and incorporates gating mechanisms to enhance both the sequence and memory representations. We introduced three regularizers that target improvements at various levels of granularity—token, windows, and sequences. Our comprehensive evaluation showed that our method not only effectively processes these time-series sequences but also surpasses the performance of existing approaches in most cases.

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

**Table A.1:** Datasets used for self-supervised classification

| Dataset | # Train | # Test | Length | # Channel | # Class |
|---|---|---|---|---|---|
| HAR | 7 352 | 2 947 | 128 | 9 | 6 |
| Epilepsy | 9 200 | 2 300 | 178 | 1 | 2 |
| Wafer | 1 000 | 6 174 | 152 | 1 | 2 |
| FordA | 1 320 | 3 601 | 500 | 1 | 2 |
| FordB | 3 636 | 810 | 500 | 1 | 2 |
| POC | 1 800 | 858 | 80 | 1 | 2 |
| ElectricDevices | 8 926 | 7 711 | 96 | 1 | 7 |

## A  DATASET SUMMARY

In our experiments, we rely on a set of well-established time-series benchmarks drawn from the UCR, UEA, and UCI repositories (Bagnall et al., 2018; Dau et al., 2018). The Human Activity Recognition (**HAR**) dataset (Anguita et al., 2013) contains triaxial accelerometer and gyroscope streams recorded at 50 Hz from 30 volunteers who carried a smartphone on their waist while performing seven everyday actions (walking, climbing or descending stairs, sitting, standing, and lying) (Yue et al., 2022). For epileptic-seizure detection we adopt the version of the **Epilepsy** dataset simplified by TS-TCC: the original EEG collection—23.6-second segments from 500 subjects and five classes (Andrzejak et al., 2001)—is reduced to a binary seizure/non-seizure task.

The remaining benchmarks—Wafer, FordA, FordB, PhalangesOutlinesCorrect (POC), and ElectricDevices—are sourced from the UCR archive (Dau et al., 2018). **Wafer** contains inline process-control sensor traces from silicon-wafer fabrication and is strongly imbalanced: defective wafers constitute 10.7% of the training set and 12.1% of the test set. **FordA** and **FordB** each comprise 500-sample engine-vibration sequences used to decide whether a specific subsystem fault is present; FordA was recorded under controlled laboratory noise, whereas FordB reflects normal operating conditions. The **POC** dataset merges three tasks derived from more than 1,300 radiographs employed for bone-age estimation, with labels indicating whether the automatically extracted phalange outlines are correct. Finally, the **ElectricDevices** dataset contains electricity-consumption profiles from 251 UK households, gathered to study residential usage patterns and help lower carbon emissions. Detailed statistics for all datasets appear in Table A.1.

## B  RECEPTIVE-FIELD GROWTH IN PROGRESSIVE MEMORY ATTENTION

**Notation.** Let $W$ be the *window length* (tokens per window), $S$ the *stride* ($1 \leq S \leq W$), $B$ the number of stacked PMA blocks, and $w \in \{0, 1, \dots\}$ the index of window $W_w$, which covers tokens $x_{wS:wS+W-1}^{\text{token}}$. Define

$$\mathcal{R}_w^{(b)} = \text{all input tokens that can influence } \textit{any} \text{ token in } W_w \text{ after block } b.$$

We assume forward (causal) processing; bidirectional results follow by symmetry.

**Lemma B.1** (base case, $b = 1$).
$$\left|\mathcal{R}_w^{(1)}\right| = W, \quad \mathcal{R}_w^{(1)} = \left[wS, wS + W - 1\right]. \tag{B.1}$$

**Lemma B.2** (without horizontal memory). *If the horizontal memory bank is* disabled*, each extra block enlarges the receptive field by $W - S$ tokens, but only until earlier windows run out:*
$$\left.\left|\mathcal{R}_w^{(b)}\right|\right|_{no\ mem} = W + (b-1)(W - S) + \min\{w, b-1\}\, S \tag{B.2}$$

**Lemma B.3** (with horizontal memory). *With the memory bank active, window $W_w$ already sees its $w$ predecessors after the* first *block:*
$$\left.\left|\mathcal{R}_w^{(b)}\right|\right|_{mem} = W + wS + (b-1)(W - S) \tag{B.3}$$

**Corollary B.3.1** (bidirectional wrapper). *Combining the forward and reversed passes yields*
$$\left.\left|\mathcal{R}_w^{(b)}\right|\right|_{bi,no\ mem} = 2\left.\left|\mathcal{R}_w^{(b)}\right|\right|_{no\ mem} - 1, \tag{B.4}$$
$$\left.\left|\mathcal{R}_w^{(b)}\right|\right|_{bi,mem} = 2\left.\left|\mathcal{R}_w^{(b)}\right|\right|_{mem} - 1. \tag{B.5}$$

*(The "$-1$" avoids double-counting the center token.)*

```python
def rf_no_mem(mW, S, B, w):
    return mW + (B-1)*(mW-S) + min(w, B-1)*S

def rf_mem(mW, S, B, w):
    return mW + w*S + (B-1)*(mW-S)

def rf_bi_no_mem(mW, S, B, w):
    return 2*rf_no_mem(mW, S, B, w) - 1

def rf_bi_mem(mW, S, B, w):
    return 2*rf_mem(mW, S, B, w) - 1
```

**Listing B.1:** Helper functions for Eqs. (B.3)–(B.5).

**FordA (UCR) example.** Series length $T = 500$, $W = 100$, $S = 25$, $B = 4$. For the last window $(w = \lfloor (T-1)/S \rfloor = 19)$, Eq. (B.3) gives

$$|\mathcal{R}_{19}^{(4)}|_{\text{mem}} = 100 + 19 \times 25 + 3 \times 75 = 500, \tag{B.6}$$

i.e., the whole series is visible after four PMA blocks.

**Remark.** After block 1 the memory bank already aggregates all $w$ previous windows, so the *token-level* receptive field is $W + wS$; subsequent blocks expand it by $(W - S)$ per layer.

## C   IMPLEMENTATION DETAILS

We train the PMT using an AdamW (Loshchilov & Hutter, 2017) optimizer with a cosine annealing learning rate (Loshchilov & Hutter, 2016) scheduler, with a warmup period of 5% to a peak of $1e-4$ and a minimum learning rate of $1e-6$. Models were trained with a batch size of 256. The models were trained on Nvidia A100 and AMD MI250x GPUs. For dataset specific hyperparameters used for Table 1, we refer the reader to the attached repository (link on page 1).

The figures seen in Section 3.3 were all created using per dataset specific frozen backbone. For both the HAR and Epilepsy figures, the backbone consisted of 6 PMA blocks with a window size of 6 and stride 3, using 2 memory states. Commonly for all visualizations, we do a simple forward pass through the model and extract the output token representations and the final PMA block's memory states. We use the mean per window memory state for the visualizations. Both the averaged memory state and the output tokens are $\ell_2$-normalized.

The scatter plots, such as Fig. J.4 use PCA for dimensionality reduction per token and per memory state. Heat-maps, such as Figs. 4 and 5 use the cosine-similarity matrix for both the token and memory state visualizations.

**Patchified input.** Our sequence encoder is a 1-D convolution with kernel $k$ and stride $k$ first tokenizes the waveform into fixed-length patch embeddings; these tokens—not the raw samples—form the input to every PMA block. Although we restrict ourselves to this patchified view in the present work, extending PMA to operate directly on raw time steps is a promising avenue for future research.

**Overlap-aware processing.** We unfold the patch stream into windows of length $W$ and stride $S$ (overlap $O = \lceil W/S \rceil$). Each window passes through a single-layer Transformer with the asymmetric mask described in Section 2.1; FlashAttention-2 reduces its space requirement to $\mathcal{O}\big((|M| + S)D\big)$ per window. After all $N = \lceil L/S \rceil$ windows are processed, the attentive overlap-pooler merges the $O$ overlapping rows at every position. Without this overlap-pooler each subsequent PMA block would receive a growing number of tokens due to $W > S$. The overlap-pooler ensures the input and output shape of any PMA block remains equal.

**Global aggregation** Finally, we append the `[CLS]` token to the output from the PMA blocks, and pass the tokens through a series of locally masked (see 2.2.2) encoder blocks to aggregate the global representation in the `[CLS]` token.

---

**Algorithm D.1: Forward pass through a $B$-block PMA stack**

**Input:** patch tokens $W_{0,1:J}$, random memory $M_r$

**Output:** token stream $\hat{W}_{B,1:J}$, memories $M_{B,1:J}$

$\hat{W}_{0,j} \leftarrow W_{0,j}, \ M_{0,j} \leftarrow \varnothing \ (\forall j)$

**for** $i \leftarrow 1$ **to** $B$ **do**

    **for** $j \leftarrow 1$ **to** $J$ **do**

        $W_{i,j}^{\text{in}} \leftarrow \sigma_{\text{residual}}(W_{0,j}, \hat{W}_{i-1,j}); M_{i,j}^{\text{init}} \leftarrow \sigma_{\text{memory}}(\text{LN}(M_{i-1,j}), \text{LN}(M_r));$

        $[M_{i,j}, W_{i,j}^{\text{out}}] \leftarrow \text{SelfAttn}([M_{i,j-1} \| M_{i,j}^{\text{init}} \| W_{i,j}^{\text{in}}]);$

    $\hat{W}_{i,j} \leftarrow \text{OverlapPool}(\{W_{i,*}^{\text{out}}\}) \ (\forall j)$

---

- $\sigma_{\text{residual}}$ — mixes new patch evidence with prior tokens.
- $\sigma_{\text{memory}}$ — gate blending inherited memory with $M_r$.
- SelfAttn — masked attention over $[M_{i,j-1} \| M_{i,j}^{\text{init}} \| W_{i,j}^{\text{in}}]$.
- OverlapPool — attentive overlap-pooler producing $\hat{W}_{i,j}$.

## D  Computational Resources

We ran the experiments on NVIDIA A100 80GB and AMD MI250x 64 GB gpus. The model can train on a single MI250x die or A100, but for higher training speed we primarily used DDP using 2 GPUs. We set number of workers per GPU to 8. The duration of a full 500 epoch experiment (as used in our experiments) using 2 GPUs is typically 7 hours. The relatively small datasets do not require much RAM, and we use 80GB RAM, yielding a healthy capacity buffer.

## E  Computational Analysis

We report minimal, reproducible compute measurements for completeness. These results detail and resource notes already in Apps. C and D and use the same backbone as the experiments. The goal is to characterize the overhead of PMA relative to a vanilla transformer with FlashAttention in the *same* implementation, rather than to claim PMT is more efficient than long-context architectures such as Transformer-XL (Dai et al., 2019) or patch-based forecasters like PatchTST (Nie et al., 2023). We distinguish between a *streaming* PMA microbenchmark (single-window kernel; no re-encoding the past) and a *vectorized* non-streaming configuration (overlapped windows materialized for speed during training).

**Hardware/precision.** Unless otherwise stated: single NVIDIA A100 (80 GB), FP16, PyTorch with FlashAttention-2 for attention kernels. Peak GPU memory is measured via `torch.cuda.max_memory_allocated()` after `cudaDeviceSynchronize()`. On shorter sequences, PyTorch's caching allocator may report the same peak for different models because blocks are reserved in advance; we report the measured peak in all cases.

### E.1  Streaming PMA microbenchmarks

We time a single PMA block with $d = 320$ (as in the main experiments), a 16 sample-convolutional patchifier ($8\times$ compression), and window stride $S = 0.5\,W$. For comparability, FlashAttention (FA) is run on the full sequence (global receptive field). We report per-window latency (ms), total sequence latency (s) implied by sliding over the sequence, maximum sustained sampling rate (Hz) with a 20% head-room, and relative FLOPs/memory versus FA (rounded).

Table E.1 show that for short sequences the writable memory trades extra compute for fixed per-step latency, but as the sequence grows the FA baseline's global attention dominates. Streaming PMA keeps the per-step memory bounded by $O((|M| + S)D)$ (App. C), which avoids the growth of full-sequence attention.

### E.2  Vectorized non-streaming inference

For training speed, we also report a non-streaming vectorized configuration that materializes all overlapped windows before the attentive overlap-pooler. All models use $d{=}320$, 6 blocks (PMT

**Table E.1:** Streaming PMA: single-window kernel timed; FA run on the full sequence. For long horizons, PMA reduces both FLOPs and peak memory while sustaining 96 kHz real-time with a 20% head-room.

| Sequence (samples → tokens) | PMA window ms (seq s) | FA full seq s | Max SR [Hz] ↑ | Δ FLOPs (PMA/FA) | | Δ mem (PMA/FA) | |
|---|---|---|---|---|---|---|---|
| 512 → 63 | 0.8 (0.01) | 0.000 | 48,000 | +95% | (0.16/0.08 G) | 0% | (26/26 MB) |
| 1,024 → 127 | 0.8 (0.01) | 0.000 | 48,000 | +82% | (0.30/0.17 G) | 0% | (26/26 MB) |
| 2,048 → 255 | 0.8 (0.01) | 0.000 | 96,000 | +70% | (0.60/0.35 G) | 0% | (26/26 MB) |
| 4,096 → 511 | 0.8 (0.01) | 0.000 | 96,000 | +53% | (1.21/0.80 G) | 0% | (28/28 MB) |
| 8,192 → 1,023 | 0.8 (0.01) | 0.000 | 96,000 | +31% | (2.52/1.93 G) | −36% | (28/44 MB) |
| 16,384 → 2,047 | 0.8 (0.01) | 0.001 | 96,000 | +6% | (5.52/5.20 G) | −68% | (30/94 MB) |
| 32,768 → 4,095 | 0.8 (0.01) | 0.003 | 96,000 | −18% | (13.0/15.8 G) | −83% | (52/308 MB) |
| 65,536 → 8,191 | 0.8 (0.01) | 0.008 | 96,000 | −37% | (33.6/53.0 G) | −95% | (52/1,092 MB) |
| 131,072 → 16,383 | 1.1 (0.01) | 0.030 | 96,000 | −49% | (98.1/191.9 G) | −97% | (122/4,262 MB) |

includes two neighborhood encoders for `[CLS]`), 100,000 time steps ($8\times$ tokenization), $W{=}0.1L$, $S{=}0.5W$, one memory token. Latency is end-to-end.

**Table E.2:** Non-streaming inference (vectorized). Vectorization duplicates overlapped windows for speed, inflating peak memory; the attentive overlap-pooler removes duplicates post-block.

| Model | Params [M] | GFLOPs | Peak mem [MB] | Latency [ms] ↓ | Tokens/s ↑ |
|---|---|---|---|---|---|
| Vanilla Transformer | 7.4 | 692.34 | 8,485 | 4,619.13 | 86,590 |
| FlashAttention (FA) | 7.4 | 692.34 | 8,336 | 3,476.55 | 115,047 |
| PMA (vectorized) | 16.3 | 329.95 | 12,292 | 2,962.21 | 135,022 |
| PMT (full) | 20.1 | 329.99 | 12,337 | 5,381.27 | 74,326 |

**Notes and caveats.** (i) FA sees the full sequence at once whereas streaming PMA strictly limits the receptive field to the current window plus memory slots; this explains FA's lower latency on short sequences and PMA's memory advantage on long horizons. (ii) The vectorized PMA/PMT inflate peak memory due to overlapped materialization; the streaming kernel avoids this by construction. (iii) Reported Δ values use rounded base numbers; minor rounding mismatch may occur.

**Reproducibility.** We use the same tokenizer, windowing, masking and attentive overlap aggregation as in the main model.

### E.3 ATTENTIVE OVERLAP AGGREGATOR COST

To understand the computational distribution within PMA blocks, we isolate and measure the *attentive overlap-pooler* component in both streaming and vectorized configurations. This component isolation gives insight into the PMA mechanisms.

**Methodology.** We measure the aggregator in isolation by providing pre-computed window embeddings, thus excluding the window encoder (PMA attention) costs. Setup matches App. E.1: A100 (80 GB), FP16, $d{=}320$, $S{=}0.5\,W$, one memory token. We report computational cost in FLOPs and the aggregator's share of total per-window processing time.

**Results.** Tables E.3 to E.5 show that the overlap aggregator accounts for only 1–3% of per-window processing time in streaming mode and less than 1% in vectorized mode. The aggregator's computational cost is dominated by key-value projections (87% of aggregator FLOPs), while the attention mechanism itself requires minimal computation due to single-token queries over $O{=}2$ positions.

**Implementation notes.** Our analysis isolates the aggregator component to measure its inherent computational cost. In production streaming systems, windows would be processed individually with aggregation happening asynchronously, avoiding the simulation overhead present in our experimental framework. The vectorized implementation materializes overlapped windows for training speed but increases peak memory; the aggregator itself contributes negligibly to this memory overhead.

**Table E.3:** Streaming overlap aggregator: isolated component analysis showing minimal overhead.

| $W$ (tokens) | $S/W$ | $O$ | $d$ | GFLOPs/step | Time share [%] ↓ |
|---|---|---|---|---|---|
| 1024 | 0.5 | 2 | 320 | 0.24 | 3.1 |
| 2048 | 0.5 | 2 | 320 | 0.48 | 1.7 |
| 4096 | 0.5 | 2 | 320 | 0.96 | 1.1 |

**Table E.4:** Vectorized overlap aggregator: computational cost scales linearly with sequence length.

| $L$ (tokens) | $W$ | $S/W$ | $K$ | $d$ | GFLOPs | MFLOPs/token |
|---|---|---|---|---|---|---|
| 25,000 | 2,500 | 0.5 | 2 | 320 | 11.7 | 0.47 |
| 50,000 | 5,000 | 0.5 | 2 | 320 | 23.4 | 0.47 |
| 100,000 | 10,000 | 0.5 | 2 | 320 | 46.8 | 0.47 |

**Table E.5:** Aggregator attribution: fraction of end-to-end time attributable to overlap aggregation.

| Configuration | Aggregator time share [%] ↓ |
|---|---|
| Streaming ($W$=2048, $S/W$=0.5, $d$=320) | 1.7 |
| Vectorized ($L$=100k, $W$=0.1$L$, $S/W$=0.5) | <1 |

## F  SUPERVISED RESULTS AND ADDITIONAL ABLATIONS

To verify that the architecture is not limited to SSL, we train PMT end-to-end with *full supervision* (cross-entropy on the label) using the same tokenizer and encoder backbone as in the main text. The `[CLS]` token is passed to a linear classification head. Unless noted, optimization and scheduling follow App. C (AdamW + cosine decay). No self-supervised losses are used in this section.

### F.1  FULL-SUPERVISION (CROSS-ENTROPY) RESULTS

Table F.1 reports Top-1 accuracy and macro-F1 on four representative datasets. Results show that PMT matches or exceeds a vanilla Transformer trained with the same supervised protocol on three of the four datasets (notably +6.0pp on FordA), with a small drop on FordB. These results confirm the *task-agnostic* nature of the backbone: while the main paper focuses on low-label SSL, the same architecture trains effectively in a purely supervised regime.

**Table F.1: Fully supervised results** (cross-entropy). Top-1 / macro-F1 (%).

| Dataset | PMT (supervised) | | Vanilla Transformer (supervised) | |
|---|---|---|---|---|
| | Top-1 | MF1 | Top-1 | MF1 |
| HAR | 97.8 | 98.0 | 97.7 | 97.9 |
| FordA | 91.5 | 91.5 | 85.5 | 85.5 |
| FordB | 79.3 | 79.2 | 80.5 | 80.5 |
| Wafer | 99.8 | 99.5 | 99.5 | 98.7 |

### F.2  SHORT-RUN ARCHITECTURAL ABLATION (HAR)

To make the architectural comparison visible in the main body while keeping training time modest, we include a 50-epoch ablation on HAR that contrasts PMT with a vanilla Transformer and two windowed variants without the full PMA mechanism. For completeness we also include the 5% SSL condition (same data, identical backbone depth/width; SSL losses only in that column). See Table F.2.

**Takeaways.** (i) In the short-run supervised setting, PMT is on par (within noise) with a vanilla Transformer. (ii) Under SSL, PMT yields consistently stronger features at 5% labels, suggesting the writable memory and progressive aggregation surface useful mid-range cues even when labels are scarce.

### F.3  ROBUSTNESS TO RESET-STATE INITIALIZATION

The memory reset token $M_r$ initializes the first-window state and can be mixed in by the learned reset gate when regimes change. We tested robustness to the random initialization of $M_r$ by repeating HAR training five times with different seeds; Table F.3 shows negligible variance.

In addition to the small numeric spread, predictions are identical for ∼99.7% of samples across seeds, indicating that PMT is insensitive to the particular initialization of the reset memory.

**Table F.2: HAR (50 epochs).** Supervised vs. 5% SSL. Top-1 / macro-F1 (%).

| Model | Supervised (CE) | | SSL (5% labels, linear probe) | |
|---|---|---|---|---|
| | Top-1 | MF1 | Top-1 | MF1 |
| PMT | 97.1 | 97.4 | 93.6 | 93.9 |
| Vanilla Transformer | 97.3 | 97.6 | 91.9 | 92.0 |
| Windowed Transformer (with temporal state pass) | 96.7 | 97.0 | 93.1 | 93.3 |
| Windowed Transformer (no state pass) | 96.8 | 97.1 | 93.0 | 93.2 |

**Table F.3: Reset-state robustness (HAR).** Supervised training with different random seeds for $M_r$; Top-1 accuracy (%).

| Seed 42 | Seed 123 | Seed 456 | Seed 789 | Seed 2024 | Mean $\pm$ Std |
|---|---|---|---|---|---|
| 96.13 | 96.19 | 96.06 | 95.99 | 95.99 | $96.07 \pm 0.08$ |

## F.4 PMA WINDOW-SIZE ABLATION

**Setup.** We study the sensitivity of PMT to the *window length* $W$ in Progressive Memory Attention (PMA). We parameterize $W$ as a fraction $\rho$ of the token length $K$ (after patchification), i.e., $W = \lfloor \rho K \rfloor$ with $\rho \in \{0.10, 0.20, 0.30, 0.50, 1.00\}$. Unless noted, we keep the stride $S = \lfloor 0.5\,W \rfloor$, use a single memory token per window ($n_m{=}1$), and leave every non-geometric hyperparameter unchanged (optimizer, temperatures for ICL/PCL/HGCL, Gaussian widths, augmentations). We pretrain on HAR with the same recipe as in the main experiments but in a short-run setting (reduced training budget); we then train linear SVM probes on the 1% and 5% label splits and report the average Top1 accuracy and macroF1. Results are averaged across the same random seeds used for our ablations.

**Table F.4: PMA window-size ablation on HAR** (short-run pretraining). $W = \rho K$, $S = 0.5W$, $n_m{=}1$. We report the average over 1% and 5% label splits. Performance is flat for $\rho \in [0.1, 0.3]$ and degrades as $W$ approaches full context.

| Window fraction $\rho$ | Avg. Top1 (%) | Avg. MacroF1 (%) |
|---|---|---|
| 0.10 | 92.8 | 93.0 |
| 0.20 | 92.7 | 93.0 |
| 0.30 | 92.7 | 92.9 |
| 0.50 | 92.5 | 92.6 |
| 1.00 | 92.3 | 92.5 |

**Wide plateau at small/mid windows.** Performance is essentially constant for $\rho \in [0.1, 0.3]$, indicating that PMAs writable memory compensates for smaller windows by accumulating context progressively across windows and depth; this aligns with the receptive-field growth in App. B, Eq. (B.3). (2) **Very large windows are unnecessary.** As $\rho \to 1.0$, accuracy and macroF1 decline slightly. With few very large windows, overlap reduces and the progressive mechanism has fewer horizontal updates, dampening the benefits of memory refresh. (3) **Practical choice.** Any $\rho$ in $[0.1, 0.3]$ is a safe default. We use $\rho{=}0.2$ in our configs to balance throughput (smaller attention tiles) and robust downstream accuracy without additional tuning. For this ablation we did *not* retune contrastive temperatures or Gaussian widths; the flat response in $[0.1, 0.3]$ therefore represents a conservative estimate.

## G ATTENTIVE OVERLAP AGGREGATOR IMPLEMENTATION DETAILS

For global position $t$ with overlapped embeddings $\{W_t^{(r)}\}_{r=1}^{O_t}$, the aggregator uses a *single query per position* to attend over overlaps and produce

$$A_t = \gamma\, \mathrm{LN}\Big(O^{-1/2}\, \mathrm{Concat}_{h=1}^{H} \sum_{r=1}^{O_t} \alpha_{h,r}\, V_{h,r}\Big), \quad \alpha_{h,r} = \mathrm{softmax}_r\Big(\tfrac{\langle q_h, K_{h,r}\rangle}{\sqrt{d_h}} + b_{h,r}\Big).$$

Here $K_{h,r}, V_{h,r}$ are per-head projections of the overlapped $W_t^{(r)}$, $q_h$ is formed by depthwise Conv1D across overlaps, mean, then a 2-layer MLP, $b_{h,r}$ is a learned head $\times$ overlap bias, and there is *no*

output projection after head concatenation. The skip path is the masked mean

$$S_t = \tfrac{1}{O_t} \sum_{r=1}^{O_t} W_t^{(r)},$$

and the block output mixes them via a learnable SkipGate:

$$\widehat{W}_t \;=\; G_{\text{skip}}(S_t,\, A_t).$$

(Streaming computes the same weighted sum without materializing $[B, L, O, D]$ and uses a tile-constant $O$ for the $O^{-1/2}$ scaling.)

## H  xLSTM AND TRANSFORMER-XL EXPERIMENT

To address the relationship between PMT's progressive memory architecture and recurrent / cache-based alternatives, we compare against xLSTM (Beck et al., 2024) and Transformer-XL (Dai et al., 2019). For both baselines we keep the tokenizer, encoder, and self-supervised objectives unchanged and simply replace the PMA stack: in one case with xLSTM layers, and in the other with a standard Transformer-XL implementation from Hugging Face that maintains a segment-level cache. Table H.1 presents results averaged across the 1% and 5% labeled data settings.

**Table H.1:** Comparison with xLSTM and Transformer-XL(averaged across 1% and 5% labeled data).

| Dataset | PMT | | xLSTM | | Transformer-XL | |
|---|---|---|---|---|---|---|
| | Top-1 Acc | Macro F1 | Top-1 Acc | Macro F1 | Top-1 Acc | Macro F1 |
| HAR | **94.2** | **94.6** | 93.5 | 93.8 | 93.0 | 93.3 |
| Epilepsy | **96.6** | **94.7** | 94.5 | 91.9 | 94.5 | 92.0 |
| Wafer | **99.1** | **97.5** | 93.4 | 81.1 | 95.7 | 92.0 |
| FordA | **88.9** | **88.9** | 88.2 | 88.2 | 88.3 | 88.3 |
| FordB | **76.5** | **76.5** | 75.5 | 75.4 | 75.6 | 75.5 |
| POC | **71.5** | 56.2 | 65.5 | **64.8** | 65.6 | 61.7 |
| ElectricDevices | 62.1 | 55.5 | 55.1 | 50.7 | 61.7 | **55.6** |
| **Average** | **84.1** | **80.6** | 80.8 | 78.0 | 82.1 | 79.8 |

All models were trained for identical epochs using the same self-supervised framework. However, neither xLSTM nor Transformer-XL exposes explicit memory state tokens, so they did not leverage our memory-aware contrastive loss (PCL) and are trained only with HGCL and ICL. PMT achieves higher average accuracy (84.1% vs 80.8% for xLSTM and 82.1% for Transformer-XL) and macro F1 (80.6% vs 78.0% and 79.8%) across the benchmark. The recurrent baselines remain competitive on some datasets (notably FordA/B and ElectricDevices), but the Transformer-XL variant can be viewed as a read-only cache analogue of PMT: past activations are reused by each window but never updated, in contrast to the writable, window-aligned memory tokens in PMA. Taken together, these comparisons illustrate how PMTs explicit writable memory and progressive attention offer a different mechanism for integrating long-range context than standard RNNs or Transformer-XL-style segment recurrence.

## I  LIMITATIONS

Our approach relies on a patchified 1-D convolutional tokenizer with fixed patch size and stride per dataset. As the patch/stride ablation in Section 3.4.1 shows, coarse downsampling (large strides) can under-sample short-lived events, and even with small stride the choice of patch size can still affect performance on datasets such as FordA/B and ElectricDevices.

A second limitation is the scope of our empirical evaluation: we focus on low-label time-series *classification* on seven UCR/UEA/UCI benchmarks, and do not evaluate PMT on other downstream tasks such as forecasting, anomaly detection, or cross-domain transfer. Finally, our contrastive losses depend on large in-batch negative sets, especially for the ICL objective, which encourages the use of batch sizes larger than those in some related work and makes it difficult to apply the same recipe to very small datasets or extremely resource-constrained hardware.

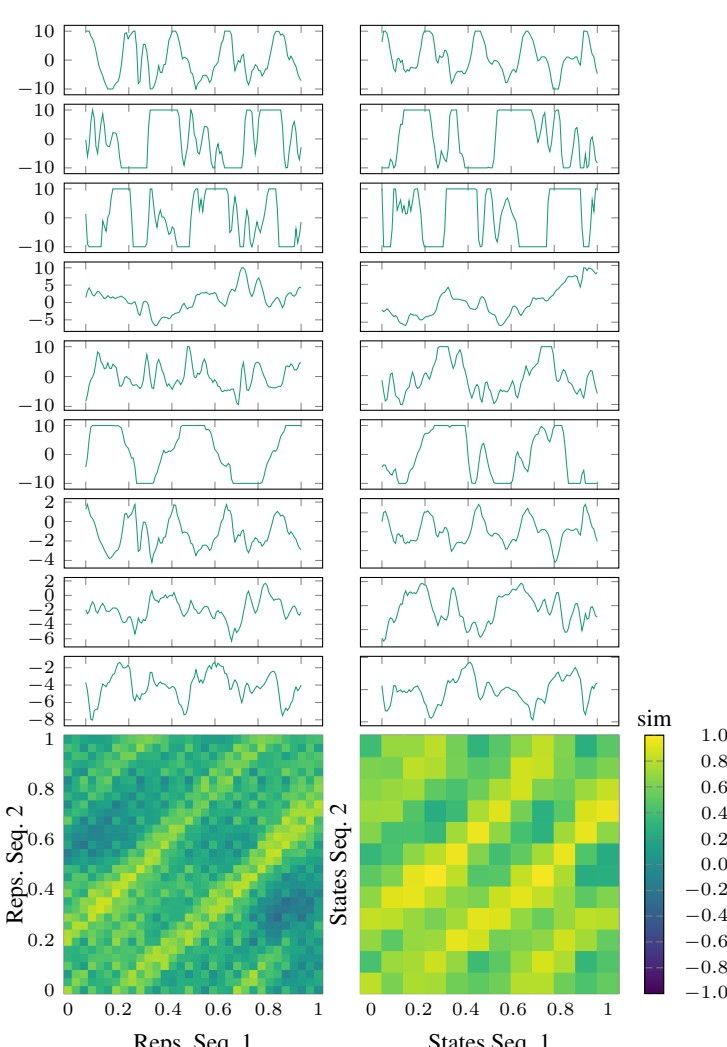

**Figure J.1:** The cosine similarity matrix between the token (bottom left) and the memory states (bottom right) representations of two signals (number 6 and 12, respectively, from the HAR validation set) of the same class (walking upstairs). The nine channels of the signals are plotted independently (top). The similarities show that the representations and states correlate to each other between the temporal domain.

## J  SUPPLEMENTARY FIGURES OF THE LEARNED REPRESENTATIONS

Figure J.1 show to spliced waveforms of the same class (walking upstairs). It shows 4, off-center, diagonal lines, together signifying the foot strikes of the subject. We included this supplementary figure for completeness, here plotting all 9 channels, rather than the single channel we plot for the other figures. Additionally, this same class comparison highlights similar patterns to what we see across all walking related classes; the foot strike pattern (diagonal lines). See Fig. J.2 and Fig. 4 for more examples of this pattern.

Figure J.3 show a double spliced waveform in the class pattern: Standing, Laying, Standing. Here we see a clear checkerboard pattern in the hybrid-to-hybrid comparison. This shows the models features produces similar features for the two unique standing waveforms, and different features for the laying class.

Figure J.2 show a 3-way semantically close but unique class waveforms. What we want to see in this figure is no correlation across splice borders. The figure shows that both the memory state and token level representations are distinct, meaning the model has learned both local-level and

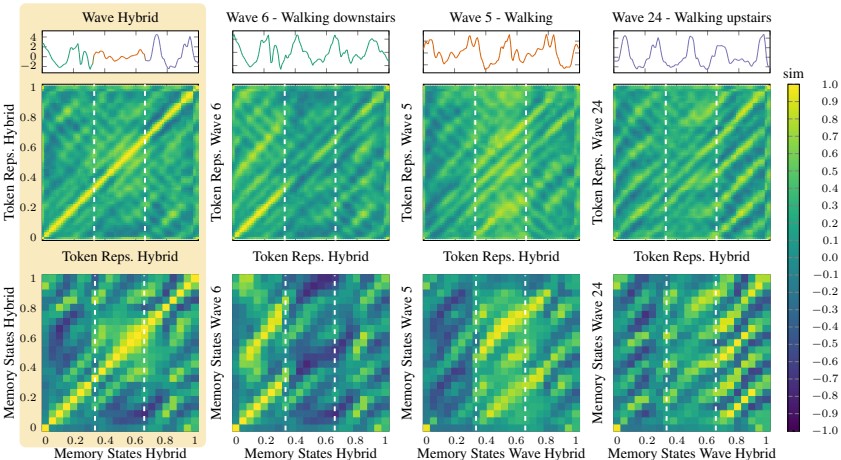

**Figure J.2:** Cosine similarity matrices for the representations and states between pair-wise signals from HAR. Higher similarity shows that the signals correlate as evidenced by the learned embeddings. The vertical bars denote the different sections of the hybrid wave. The wave number corresponds to the sample index in the validation dataset.

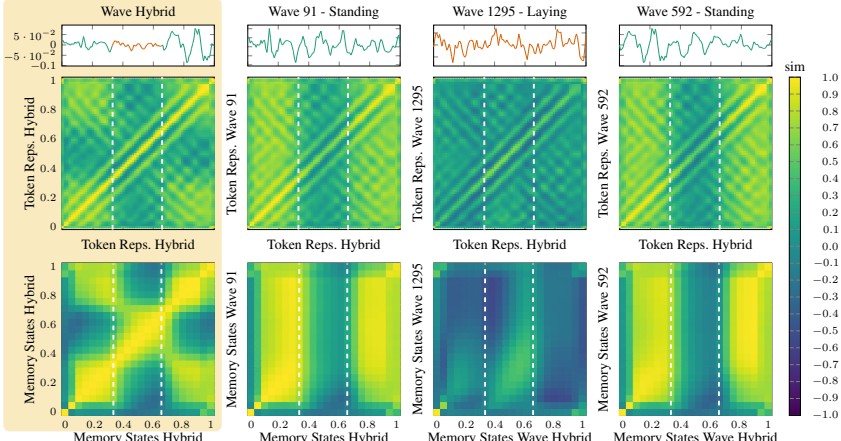

**Figure J.3:** Cosine similarity matrices for the representations and states between pair-wise signals from HAR. Higher similarity shows that the signals correlate as evidenced by the learned embeddings. The vertical bars denote the different sections of the hybrid wave. The wave number corresponds to the sample index in the validation dataset.

medium-range *class-separable* and *temporal compositional* representations for these semantically proximal activities.

Figure J.4 shows the token representations, and the memory states over a waveform from HAR spliced from 2 waveforms of different classes, walking (first and last third) and walking downstairs (see Appendix C for details). We plot only one channel of the 9 here for interpretability, while the model processed all. As the PMA unrolls over windows of tokens, with a stride greater than one, we have more token representations (*local nuance*), than memory states (*mid-range motifs*). The center figure highlights the ability of PMT to extract token representations that are semantically consistent with the underlying signal. This is observed by the linear separability of the tokens based on their source. This observation extends to the memory-state Fig. 4 (however, we note that the first memory-state in the sequence is often less semantically significant due to its small effective receptive field, see Appendix B.

Finally, Fig. J.5 shows Ts2Vec + SoftCLTs embedding at layer 5 and the final layer, as trained in the softCLT paper (Lee et al., 2024). This figure is using the same data splice we use in Fig. 4.Ts2vec's

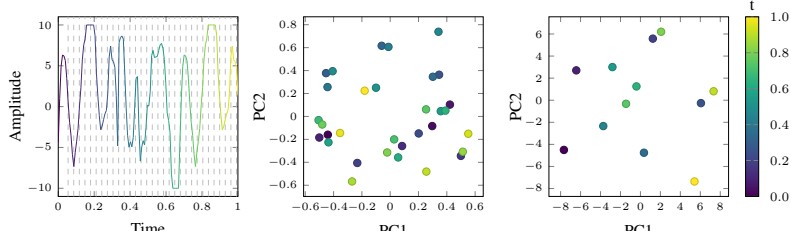

**Figure J.4:** (Left) Double spliced HAR waveform. (Middle) PCA of representations (tokens) of the signal. (Right) PCA of memory-states.

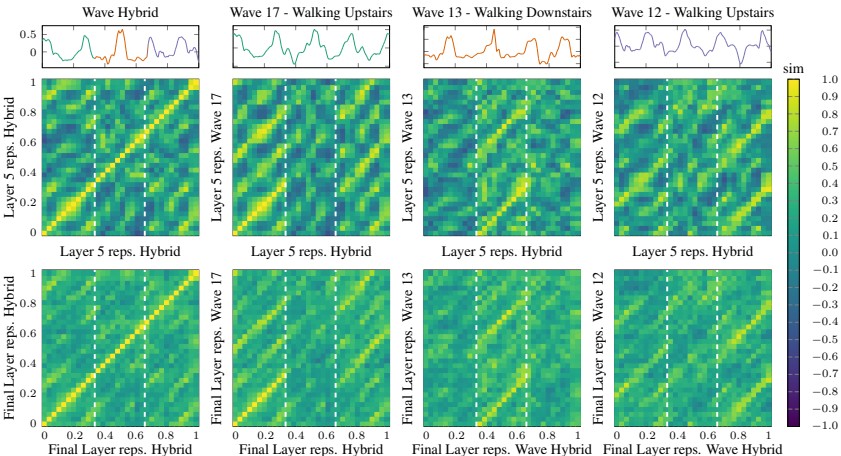

**Figure J.5:** Cosine similarity matrices for the embeddings from layer 5 and 10 (final) from Ts2Vec + Softclt between pair-wise signals from HAR. Higher similarity shows that the signals correlate as evidenced by the learned embeddings. The vertical bars denote the different sections of the hybrid wave. The wave number corresponds to the sample index in the validation dataset. Same samples as Fig. 4.

stacked dilated convolution layers extract the "foot-strike" pattern we see in Fig. 4, but lacks the "checkerboard" pattern in the hybrid-to-hybrid comparison. We included this figure for completeness, despite the extraction at different layers approach, may not be perfectly analogous to the token vs. memory-state comparison.

# K  LLM DISCLOSURE

We used a large language model as a writing assistant to improve clarity and grammar, and to help surface potentially relevant related work during scoping. All technical claims, modeling choices, experiments, analysis were the work of the authors. No citations were included without verification. No empirical results were generated with LLMs.

