# OpenReview forum: "Progressive Memory Transformers: Memory-Aware Attention for Time Series"
_ICLR.cc/2026/Conference — Submitted to ICLR 2026_

### Official Review · Reviewer_M7TR · 2025-10-31

**Soundness:** 2
**Presentation:** 3
**Contribution:** 2
**Rating:** 4
**Confidence:** 4

**Summary:**

The paper proposes Progressive Memory Transformer (PMT), a sliding-window Transformer architecture that maintains a writable memory bank for each window and layer. Memory states are updated via reset/carry gates and propagated across both time (window-to-window) and depth (layer-to-layer). The model is trained using three contrastive losses at different temporal scales: a hierarchical Gaussian contrastive loss (HGCL) for local structure, a progressive contrastive loss (PCL) for memory supervision, and an instance-level contrastive loss (ICL) for global alignment. Experiments on several small-scale time-series classification datasets (UCR/UEA/UCI) under low-label settings (1–5%) show moderate improvements over baselines.

**Strengths:**

- The paper is clearly written, with well-explained figures and masking diagrams.
- The proposed memory mechanism is conceptually sound: instead of read-only caches (e.g., Transformer-XL), it introduces writable, gate-controlled slots aligned to each sliding window.
- The design of contrastive losses targets local, intermediate, and global temporal structures.
- Visualization of memory activations and ablations of loss weights are helpful to understand how each component behaves.
- Computational analysis is provided.

**Weaknesses:**

- The novelty needs to be clarified. The paper mainly combines writable memory (as in Transformer-XL, Compressive Transformer, and Titans) with hierarchical contrastive objectives.

- Evaluation covers only seven small classification datasets with short sequences.

- Transfer learning under in- and cross-domain scenarios is not tested.

**Questions:**

- Could you show ablations replacing writable memory with a read-only cache (Transformer-XL style) to isolate the benefit of “writability”?

- How sensitive is the model to patch size or window length?

- Do the authors think any experiments on broader tasks are reasonable? i.e., softCLT is tested on forecasting.

---

> ### Author Response · Authors · 2025-11-25
> **Author Rebuttal (1/3)**
>
> Thank you for your thoughtful review. To help locate the revisions to the manuscript that directly address your concerns, see changes highlighted in blue. The highlighted changes which multiple reviewers asked for are pink.
>
>
> > "The novelty needs to be clarified. The paper mainly combines writable memory (as in Transformer-XL, Compressive Transformer, and Titans) with hierarchical contrastive objectives."
>
> While other reviewers describe PMT as an “architecturally novel” and “timely, well‑motivated” memory‑augmented transformer, we agree that our positioning relative to specific memory and SSL baselines could be clearer. In the revised manuscript we therefore expanded the paragraph **Positioning relative to prior work** (highlighted in blue) to make these distinctions explicit.
>
> Architecturally, PMT is more than a generic "writeable memory + SSL loss" combination. It introduces a *window-aligned, writable memory hierarchy* that is tailored to sliding-window, patchified time series, and couples this with a contrastive protocol that *directly supervises* what is written into memory at multiple temporal scales. Transformer‑XL and Compressive cache past activations in a read‑only segment memory, Perceiver‑style latents are global bottlenecks that reset every sequence, and Titans uses a persistent global slot bank detached from sliding windows. In contrast, each PMA block maintains sample‑specific memory tokens tied to the current overlapping window; they are updated by attention using both temporal and hierarchical context and mixed with a reset state via a gate designed to overwrite stale context when beneficial. Appendix B derives how this yields progressive receptive‑field growth without re‑encoding the full past.
>
> On the objective side, hierarchical time‑series SSL methods such as TS2Vec, SoftCLT, and TNC attach losses to pooled or strided token features and never supervise memory explicitly. Our HGCL preserves full patch tokens and defines Gaussian‑weighted token/window positives with in‑batch negatives, while PCL treats the writable memory tokens themselves as contrastive objects and ICL aligns [CLS] summaries.
>
> Appendix H provides a direct test: we replace PMA with a Transformer-XL segment cache under the same tokenizer and HGCL/ICL protocol. Transformer-XL reuses past activations but its cache is read-only—there is no learned "what to write" decision for PCL to supervise. PMT achieves 84.1% average Top-1 vs. 82.1% for Transformer-XL across all seven benchmarks. The read-only cache recovers part of the benefit of reusing past context, but the writable memory with direct supervision provides a further 2pp gain on average.
>
>
> > "Evaluation covers only seven small classification datasets with short sequences."
>
> Our empirical study focuses on seven widely used UCR/UEA/UCI time‑series classification benchmarks—HAR, Epilepsy, Wafer, FordA, FordB, PhalangesOutlinesCorrect (POC), and ElectricDevices—which jointly span diverse domains (human activity, EEG, manufacturing, engine vibration, radiograph quality, household electricity), channel counts (1–9), sequence lengths between 80 and 500 time steps, and class cardinalities from binary to 7‑way classification (Table A.1). FordA and FordB, in particular, provide challenging 500‑step engine‑vibration sequences whose labels depend on short‑lived phase and frequency shifts; with the improved tokenizer from Sec. 3.4.1, PMT is now competitive on both datasets in the low‑label regime (Table 1). This range of sequence lengths is typical for recent work on low‑label UCR/UEA classification, but it does mean that our experiments do not target ultra‑long horizons with thousands of time steps; we view those as a complementary evaluation setting rather than the focus of the present paper.
>
> In terms of sample size, the number of sequences in each benchmark is moderate compared to large industrial time‑series logs or massive event datasets, but together the seven training splits still contain ≈21.5M time steps—over 40% of the combined UCR+UEA volume. This makes the 1–5% label regime genuinely stringent: on several datasets, 1% of labels corresponds to only a handful of labeled sequences per class, so gains cannot be attributed to simply having millions of labeled examples but instead reflect improvements in the learned representations. Within this regime we intentionally prioritized depth over breadth, complementing the main 1%/5% linear‑probe results with loss‑component ablations, PMA window‑size ablations, the new patch/stride study on FordA/B and ElectricDevices, qualitative analyses of token and memory embeddings, supervised PMT vs. a vanilla Transformer, and replacements of PMA with xLSTM and Transformer‑XL under the same tokenizer and SSL protocol (Secs. 3.3–3.4, Apps. F–H). We now state explicitly in the Limitations section (App. I) that extending this evaluation to longer‑horizon datasets and broader task families is an important direction for future work.

---

> ### Author Response · Authors · 2025-11-25
> **Author Rebuttal (2/3)**
>
> > "Could you show ablations replacing writable memory with a read-only cache (Transformer-XL style) to isolate the benefit of 'writability'?"
>
> We agree this is an important experiment to include, and we now include this exact comparison in the revision (Appendix H, Table H.1). We replace the PMA stack with a Transformer‑XL–style segment cache while keeping everything else fixed: same convolutional tokenizer, windowing, augmentations, HGCL/ICL setup, optimization schedule, and evaluation protocol. As Transformer‑XL does not expose explicit memory tokens, this variant cannot use PCL; it is trained only with HGCL and ICL. The PMT backbone, by contrast, uses window‑aligned writable memory tokens that are updated via attention and explicitly supervised by PCL.
>
> The new results are across the seven datasets, PMT achieves 84.1% vs. 82.1% average Top‑1 accuracy and 80.6% vs. 79.8% macro‑F1 compared to the Transformer‑XL read‑only cache variant, with the largest gaps on Wafer and Epilepsy and small but consistent improvements on FordA/B and ElectricDevices. This suggests that reusing past activations via a read‑only cache already recovers part of the benefit of longer context, but progressively updating window‑aligned memory slots and supervising them with PCL provides an additional gain beyond what a static cache can offer.
>
> To further contextualize this, Appendix F, Table F.2 adds a shorter‑run architectural ablation on HAR under both full supervision and 5%‑label SSL. Here we compare PMT to (i) a vanilla Transformer, and (ii) “windowed Transformers” with and without passing any temporal state between windows. In the SSL setting, PMT reaches 93.6/93.9 Top‑1/macro‑F1, outperforming the stateless Transformer (91.9/92.0) and slightly exceeding both windowed variants (93.1/93.3 and 93.0/93.2). Together with the Transformer‑XL ablation, these experiments indicate that (i introducing any form of segment state (read‑only cache or simple state pass) is beneficial over a purely stateless encoder; and (ii) PMT’s writable, window‑aligned memory tokens plus PCL yield the strongest representations among these designs under a shared SSL pipeline.
>
> We have added pointers to these results in the main text (Sec. 3.2) and clarified explicitly that the Transformer‑XL baseline serves as a read‑only cache control for the “writability” of PMT’s memory. Changes highlighted in pink.
>
>
> > "How sensitive is the model to patch size or window length?"
>
> We agree that PMT’s behavior depends on the tokenizer and the PMA window length, and our analysis now makes this explicit. In the revision we add a targeted patch/stride ablation on FordA, FordB, and ElectricDevices (Sec. 3.4.1, Table 4), and we point the reader to the PMA window-length ablation on HAR that was already present in the original submission (App. F.4). The new patch/stride study shows that, at fixed stride, varying patch size has only a moderate effect, while performance is strongly governed by overlap: increasing the stride from heavily overlapping to non-overlapping patches causes a sharp monotonic drop on FordA/B and a smaller but consistent degradation on ElectricDevices. This confirms that our original “patchification loss” explanation was really about coarse temporal downsampling, not patch size per se. Using the small-stride configuration suggested by this ablation, we update the main tokenizer for FordA/B and obtain substantially stronger results than in the original submission, with PMT now clearly competitive with TS2Vec+SoftCLT, particularly in the 1% label regime (Table 1). We appreciate the push to explore this further.
>
> For the PMA window length, Appendix F.4 sweeps the window fraction ρ on HAR and finds a broad plateau: PMT is stable for a wide range of window sizes as long as there is some overlap, with only a mild degradation when the window approaches full-sequence length. This experiment was already included in the original appendix; in the rebuttal we simply make its role explicit when discussing sensitivity to window length. Together, these studies indicate that PMT is reasonably robust to patch size and window length, and that the main sensitivity arises from stride/overlap, which we now tune explicitly on the most challenging datasets.

---

> ### Author Response · Authors · 2025-11-25
> **Author Rebuttal (3/3)**
>
> > "Transfer learning under in- and cross-domain scenarios is not tested." and "Do the authors think any experiments on broader tasks are reasonable? i.e., softCLT is tested on forecasting."
>
> We agree that both transfer learning (in‑ and cross‑domain) and broader tasks such as forecasting, anomaly detection, and other downstream objectives are important directions for time‑series SSL, and recent work such as TS2Vec, TS‑TCC, and SoftCLT has begun to establish useful protocols for these settings. In this work, however, we intentionally scope the empirical study to low‑label classification and aim for depth in that regime rather than breadth across task families. The seven benchmarks we use cover a range of domains, channel counts, class structures, and sequence lengths up to 500 time steps (Table A.1), and within this setting we devote our experimental budget to understanding the behavior of PMT itself: loss‑component ablations, PMA window-size ablations, the patch/stride study on FordA/B and ElectricDevices, qualitative analyses of token and memory embeddings, supervised PMT vs. a vanilla Transformer, and replacements of PMA with xLSTM and Transformer‑XL under the same pipeline (Sections 3.3–3.4, Appendices F–H).
>
> Unlike methods such as SoftCLT, which can be added on top of existing encoders and evaluation setups, our contribution combines a new memory‑augmented backbone with a three‑level contrastive protocol. Fully characterizing how these two components behave already requires a substantial number of runs in a single, coherent setting; adding forecasting, anomaly detection, and transfer on top would introduce additional axes of variation (task‑specific heads, horizons, metrics, baseline recipes) and either make the paper unwieldy or force a much more superficial treatment of each task. We therefore view this paper as establishing the core architecture and loss protocol in a controlled setting, laying a foundation that later work can build on when exploring broader task families. Architecturally, PMT is task‑agnostic: the backbone produces patch‑, window‑, and sequence‑level representations, and Appendix F shows it also trains effectively in a fully supervised regime, so we see applying the same backbone to forecasting, anomaly detection, and longer‑horizon problems as a natural follow‑up project rather than something we can do rigorously within the scope of this paper. We now state this explicitly in the Limitations section (Appendix I), where we list broader tasks and longer sequences as directions for future work.

---

### Official Review · Reviewer_iV7V · 2025-10-31

**Soundness:** 3
**Presentation:** 3
**Contribution:** 3
**Rating:** 6
**Confidence:** 3

**Summary:**

The paper introduces Progressive Memory Transformer (PMT), a lightweight transformer architecture for self-supervised time-series representation learning. Unlike conventional stateless transformers, PMT maintains a writable, hierarchical memory across overlapping temporal windows, allowing representations to accumulate information progressively rather than repeatedly re-encoding past segments. In particular, the core of the proposed PMT is Progressive Memory Attention (PMA), which combines causal attention with adaptive gating and reset mechanisms, enabling selective retention, refinement, or overwriting of context. Although this design is reminiscent of an LSTM, the authors do not appear to draw any explicit connection to it. On top of this architecture, the authors propose a three-level contrastive learning framework to capture local nuance, mid-range motifs, and global semantics: i) Hierarchical Gaussian Contrastive Loss – enforces smoothness among nearby tokens and overlapping windows; ii) PMA Contrastive Loss (PCL) – supervises writable memory tokens to capture mid-range temporal motifs; iii) Instance Contrastive Loss (ICL) – aligns sequence-level `[CLS]` representations for global semantics.

**Strengths:**

- The proposed memory-augmented transformer for time-series SSL is a timely and well-motivated innovation. The proposed PMT ticks a few boxes of the memory-augmented transformers by maintaining a writable, hierarchical memory across overlapping temporal windows. The authors further propose a multi-scale contrastive loss to capture local nuance, mid-range motifs, and global semantics in one framework.
- Empirical evaluations on UCR/UEA/UCI benchmarks show promising performance, especially in the few-label regime.
- The paper is, in general, well-structured.
- The code is available.

**Weaknesses:**

- The design of PMA bears a strong resemblance to LSTM architectures. However, the authors do not appear to explicitly appreciate or elaborate this connection. Similar to LSTM’s hidden and cell states, PMT maintains a persistent memory that is progressively updated across temporal windows, enabling earlier segments to influence subsequent ones. It also incorporates learnable gating mechanisms that determine how much prior memory is retained, refined, or reset—analogous to the forget, input, and output gates in LSTMs. However, unlike LSTM, PMT appears less effective at capturing long-range dependencies, (as evidenced by its weaker performance on the Ford A/B datasets) due to its dependency on patch size.

- Following my previous point, it might also be interesting to see the ablations on different patch sizes, at least for Ford A/B datasets.
- Comparisons to prior arts, e.g., Titans, Transformer-XL, on the same setups would consolidate the paper more.
- The evaluations are limited to the classification task. What about other TS tasks, such as forecasting and anomaly detection ?
- Although “lightweight,” the hierarchical PMA still appears to be heavy. It might be more interesting to see comparisons to lightweight CNN-based SSL baselines.

**Questions:**

See weaknesses.

---

> ### Author Response · Authors · 2025-11-25
> **Rebuttal by Authors (1/2)**
>
> Thank you for your thoughtful review. To help locate the revisions to the manuscript that directly address your concerns, see changes highlighted in green. The highlighted changes which multiple reviewers asked for are pink.
>
>
> > Weakness 1 & 2: (LSTM similarity, FordA/B performance, & patch-size ablations)
>
> We thank the reviewer for the analysis. We agree that PMA's use of a persistent state with learned gates is reminiscent of LSTMs: earlier windows influence later ones, & the reset gate decides when prior memory should be retained or overwritten. This similarity is now made explicit in Section 2.1 (highlighted in green), where we note that the reset gate plays a role analogous to forget/reset gates in LSTMs, but applied to *window-level memory tokens updated via attention rather than per-timestep hidden states*. At the same time PMT also inherits structure from CNN-style patchified encoders (sliding windows over a fixed convolutional stem) & from memory-augmented transformers such as Transformer-XL & Titans. Our goal is not to present PMT as "just an LSTM", but as a **memory-augmented transformer** whose writeable, window-aligned memory tokens are explicitly supervised by a mid-range contrastive loss (PCL) on top of HGCL & ICL. We also clarify this relationship in Appendix H: the **original submission** already included an xLSTM replacement for the PMA stack to probe this recurrent connection, & in the revision we extend this experiment to additionally include Transformer-XL under the same tokenizer & SSL objectives.
>
> Following the reviewer's suggestion, we ran a targeted ablation over the tokenization step, targeting patch size & stride on FordA/B & ElectricDevices (Section 3.4.1, Table 4, highlighted in pink). At a fixed stride, varying patch size causes only moderate changes in accuracy, indicating a broad plateau once patches are sufficiently long to capture the relevant vibration patterns. In contrast, at a fixed patch size, increasing the stride from heavily overlapping to non-overlapping patches causes a sharp degradation on FordA/B & a smaller monotonic drop on ElectricDevices. This supports the interpretation that the main failure mode was coarse temporal downsampling. Leveraging the results of this ablation, we updated the main classification table & section (Table 1, Section 3.2, highlighted in pink), with the updated tokenization parameters. **PMT is now even more competitive, particularly in the 1% label setting, on both FordA & FordB.**
>
> Finally, we note that this recurrent connection was already reflected in our experimental design: the original submission included Appendix H, were compared under a shared self-supervised setup. In the revision, we update both PMT & xLSTM in that table to use the improved tokenizer from section 3.4.1, & added a Transformer-XL comparison. All three models benefit from the revised tokenizer, but PMT retains the best average Top 1 accuracy & macro-F1 & a small edge on FordA/B. We thank the reviewer for encouraging us to explore the tokenization more carefully, which led to improved performance & clearer insight into PMT’s failure modes.
>
>
> > "Comparisons to prior arts, [...] would consolidate the paper more."
>
> We agree that comparing PMT with prior memory-augmented architectures under the same self-supervised setup strengthens the paper. In the revision, we therefore add a controlled experiment where we replace the PMA stack with either xLSTM or Transformer-XL, keeping the tokenizer, augmentations, & HGCL/ICL training protocol unchanged. PMT retains the best average Top 1 Accuracy & macro-F1 across all seven benchmarks (Appendix H, Table H.1) under this shared pipeline.
>
> By contrast, Titans relies on a global persistent slot bank & test-time memorization & currently lacks a drop-in implementation for our patchified sliding-window SSL framework, so integrating it would require substantial re-engineering & additional tuning beyond the rebuttal scope. Thus, for this revision we therefore focus on xLSTM (recurrent) & Transfomer-XL (segment cache) as the most directly comparable memory-augmented baselines.

---

> ### Author Response · Authors · 2025-11-25
> **Rebuttal by Authors (2/2)**
>
> > "The evaluations are limited to the classification task [...]"
>
> We agree that restricting our evalutation to classification is a limitation. In this work we deliberately focus on low-label time-series classification on seven UCR/UEA/UCI benchmarks (HAR, Epilepsy, Wafer, FordA, FordB, POC, ElectricDevices; Section 3.1), chosen to balance domain & length diversity with sufficient unlabeled volume for in‑batch contrastive learning (∼21.5M time steps across training splits). Within this regime we **emphasize depth rather than breadth**: beyond the main 1%/5% linear‑probe results, we include loss-component ablations (Table 2 & 3), patch/stride sensitivity (Section 3.4.1, Table 4), architectural & supervised ablations (Appendix F), qualitative analyses of token/memory representations (Section 3.3, Appendix J), & replacements of PMA with xLSTM & Transformer‑XL (Appendix H).
>
> Architecturally, PMT is task‑agnostic: the backbone exposes patch tokens, memory tokens, & a [CLS] summary, & Appendix F shows that the same architecture also performs well when trained end‑to‑end with cross‑entropy. We expect these interfaces to support forecasting (e.g., predicting future patches from the current window+memory state) & anomaly detection (e.g., scoring tokens or windows), but a rigorous study would require adapting the pipeline to task‑specific metrics & re‑tuning specialized forecasting baselines, which is beyond what we can execute within this revision. We now state explicitly in the Limitations section (Appendix I, highlighted in pink) that our empirical study is restricted to low‑label classification & that extending PMT to forecasting, anomaly detection, & transfer is left to future work.
>
> > "Although 'lightweight,' the hierarchical PMA still appears to be heavy [...]"
>
> We initially wrote "lightweight" in relation to the memory footprint of a standard transformer, however, we see how this can be interpereted in a broader view. We absolutely agree that the computational footprint of PMT was under-documented for such a broad claim, & it is not our intention to misrepresent in such a way.
>
> In the revision we made two changes (i) We remove the "lightweight" claim & describe PMT simply as a memory-augmented transformer, & (ii) In appendix E, computational analysis, we make it explicit that the analysis is meant to characterize the overhead of PMA relative to a standard Transformer, & we do not claim that PMT is more efficient than, nor as optimized, as long-context architectures such as Transformer-XL or patch-based forecasters such as PatchTST. Table 1 already compares PMT to several strong CNN‑backbone baselines (SSL‑ECG, TS2Vec+SoftCLT) on the same classification tasks, where PMT is competitive or better in the low‑label regime. However, we do not perform computational comparisons as we view this beyond the scope of our work.

---

> > ### Comment · Reviewer_iV7V · 2025-11-25
> >
> > I thank the authors for the extensive and insightful response. Most of my initial concerns have been adequately addressed. The paper has been largely consolidated with the newly provided experiments and polished writing. The reviewer does not have further concerns and recommends acceptance of the paper.

---

### Official Review · Reviewer_xeyo · 2025-11-01

**Soundness:** 3
**Presentation:** 3
**Contribution:** 3
**Rating:** 6
**Confidence:** 3

**Summary:**

The paper proposes the Progressive Memory Transformer (PMT), a self-supervised architecture for time-series representation learning.
PMT introduces writable, hierarchical memory units that propagate contextual information across overlapping windows, enabling progressive context aggregation without re-encoding the entire sequence.
Three complementary contrastive objectives are defined:
(1) HGCL (Hierarchical Gaussian Contrastive Loss) – encourages local smoothness within windows.
(2) PCL (PMA Contrastive Loss) – aligns intermediate-level memory semantics.
(3) ICL (Instance Contrastive Loss) – preserves global sequence consistency.
Experiments on seven benchmarks (HAR, Epilepsy, Wafer, FordA/B, POC, ElectricDevices, etc.) show strong results, especially under low-label (1–5%) regimes.

**Strengths:**

1. Architectural novelty:
The proposed progressive memory attention (PMA) extends Transformer-XL with writable memory propagation across both temporal windows and model layers. The idea of dynamically updating memory states is original and technically sound.

2. Hierarchical learning objective:
The combination of HGCL, PCL, and ICL aligns well with the model hierarchy (token → window → sequence), achieving coherent multi-scale supervision.

3. Comprehensive evaluation:
The model is validated on seven diverse benchmarks and includes reasonable ablations on the loss components and qualitative memory visualizations.

4. Interpretability attempt:
Visualizations of middle-layer memory embeddings illustrate the progressive clustering of classes, partially supporting the claimed representational hierarchy.

**Weaknesses:**

(1). Limited and inconsistent empirical superiority.
PMT is not consistently better than strong baselines such as TS2Vec and SoftCLT.
Performance drops on datasets such as FordA, FordB, and ElectricDevices are attributed to "patchification loss," yet this claim is speculative. No patch-size or stride sensitivity analysis supports it.

(2). Relation to recent global-token approaches.
Recent work, for example "Sequence Complementor: Complementing Transformers for Time Series Forecasting with Learnable Sequences" (AAAI 2025), introduces learnable global tokens that complement local context and achieve similar goals of long-range dependency modeling.PMT’s progressive memory resembles such global or complement tokens, yet the paper does not clearly distinguish whether writable memory offers capabilities beyond static learnable tokens or Perceiver-style latent arrays.

(3). The proposed writable memories, overlap pooling, and gating introduce extra computation, but there is no quantitative comparison of FLOPs, memory footprint, or runtime versus other baseline methods (Transformer-XL or PatchTST).

**Questions:**

see weakness above

---

> ### Author Response · Authors · 2025-11-25
> **Rebuttal by Authors**
>
> Thank you for your thoughtful review. To help locate the revisions to the manuscript that directly address your concerns, see changes highlighted in orange. The highlighted changes which multiple reviewers asked for are pink.
>
>
> > "Limited and inconsistent empirical superiority... attributed to "patchification loss," yet this claim is speculative. No patch-size or stride sensitivity analysis supports it."
>
> We agree that our original discussion of “patchification loss” was under‑supported. In the revised manuscript we therefore revisit this hypothesis with a dedicated ablation over patch size and stride on FordA, FordB, and ElectricDevices (Sec. 3.4.1, Table 4). The ablation shows that the main failure mode on these benchmarks is coarse temporal downsampling: large strides under‑sample short‑lived events so that the corresponding patterns never enter the token sequence. Holding the patch length fixed and increasing the stride from 10% to 100% of the patch size reduces the average top‑1 accuracy on FordA from 87.7% to 62.5%; FordB exhibits the same monotonic trend, and ElectricDevices also degrades with increasing stride, although more mildly (61.5% to 57.9%). These results indicate that, on these datasets, performance is governed primarily by the tokenizer’s temporal overlap rather than by patch length itself, and that PMT behaves as expected for patchified encoders whose labels depend on short‑lived events.
>
> We thank the reviewer for encouraging us to investigate this aspect more carefully: the new ablation both improves our understanding of PMT’s failure modes and leads to better results. With the original tokenizer, PMT was already competitive on FordA/B in the 1%‑label regime (82.1% average top‑1 vs. 80.5% for TS2Vec+SoftCLT); adopting the small‑stride configuration suggested by this study further increases this margin to 82.9% (Table 1), and we now report all main numbers under this empirically justified tokenizer for FordA and FordB.
>
> See changes to the manuscript highlighted in pink.
>
> > "Relation to recent global-token approaches. Recent work, for example "Sequence Complementor: Complementing Transformers for Time Series Forecasting with Learnable Sequences" (AAAI 2025)... the paper does not clearly distinguish whether writable memory offers capabilities beyond static learnable tokens or Perceiver-style latent arrays."
>
> We thank the reviewer for pointing out recent work on learnable global tokens for time-series forecasting. Sequence Complementor (Chen et al., AAAI 2025) augments PatchTST-style forecasters with a small set of learnable complementary sequences that are concatenated to the patchified input and regularized via a diversification loss. These tokens are shared across all samples and act as static global latents: they interact with local tokens via self-attention during encoding but are reset for each sequence and discarded after the encoder, so they do not maintain sequence-specific state across windows. In contrast, PMT's memory tokens are *sample-specific and writable*. Each window/level carries its own memory state that is updated via causal attention from both the previous window and layer, mixed with a reset token through a learnable gate, and explicitly supervised by our PMA Contrastive Loss (PCL). This progressive, window-aligned state treats memory as a sequence-dependent representation that is propagated across windows and depth, rather rather than as static global tokens, and is designed to let PMT reuse context without repeatedly re-encoding the full past.
>
> > "The proposed writable memories, overlap pooling, and gating introduce extra computation, but there is no quantitative comparison of FLOPs, memory footprint, or runtime versus other baseline methods (Transformer-XL or PatchTST)."
>
> Our use of the word “lightweight” in the abstract was misleading here. We only meant “lightweight” relative to a vanilla transformer that repeatedly re-encodes the full past, not in the sense of being more efficient than highly optimized long-context models such as Transformer‑XL or patch‑based forecasters like PatchTST. In the revision we therefore remove this wording and describe PMT more neutrally as a memory‑augmented transformer.
>
> Appendix E already provides quantitative FLOPs, peak memory, and latency measurements comparing PMA against a matched vanilla Transformer with FlashAttention‑2 under the same tokenizer and implementation, and shows that the extra components (overlap pooling, gating, and writable memory) contribute only a small fraction of the total cost. Our goal with this analysis is to bound the overhead of the proposed mechanisms inside a single codebase, not to claim superiority over specialized architectures like Transformer‑XL or PatchTST. Including a compute analysis comparison to PatchTST and/or Transfomer-XL would absolutely make sense if this was an aspect we intended to claim competitiveness in.

---

### Author Response · Authors · 2025-12-03
**On the flaws, the fixes, and the case for PMT**

### Executive summary

We propose the Progressive Memory Transformer (PMT), a stateful, memory-augmented backbone with window‑aligned writable memory & a three‑level contrastive protocol for self‑supervised time‑series learning. PMT achieves the best average Top‑1 accuracy at 1% labels (82.9% vs 80.5% for TS2Vec+SoftCLT) across seven benchmarks.

**Reviewer stance.** Two reviewers (xeyo, iV7V; both 6) describe PMT as *novel, timely, & technically sound*. After seeing the new experiments, iV7V **explicitly recommends acceptance.** The third reviewer (M7TR; 4) finds the method *"conceptually sound"* & *"would not mind if the paper is accepted."*

### We summarize the main concerns & how the revision addresses them.

1. **Novelty & relation to prior memory/SSL work (M7TR vs xeyo & iV7V)**

    M7TR argues that PMT "mainly combines writable memory with hierarchical contrastive objectives," whereas xeyo & iV7V describe the architecture as *architecturally novel & timely*.

    In the revision we:
    - Clarify positioning vs **Transformer‑XL, Compressive Transformer, Perceiver, Titans, & global‑token approaches** in the "Positioning relative to prior work" paragraph & Section 4. PMT uses **window‑aligned, sample‑specific writable memory tokens** that propagate horizontally (window‑to‑window) & vertically (level‑to‑level), with a reset gate & **direct supervision on the memory tokens (PCL).**
    - Provide **controlled comparisons under a shared SSL pipeline** (same tokenizer, augmentations, HGCL/ICL setup) in Appendix H, replacing PMA with xLSTM & with a Transformer‑XL‑style read‑only cache:
        - Top‑1: 84.1% (PMT) vs 82.1% (Transformer‑XL) vs 80.8% (xLSTM)

    These results show that **read‑only caches & recurrent baselines recover part of the benefit of longer context, but writable, window‑aligned memory with explicit supervision yields consistent additional gains**.

2. **Patch/stride sensitivity & FordA/B performance (xeyo, iV7V, M7TR)**

    All three reviewers raised concerns about FordA/B & the role of patchification. In response we added a **patch‑size & stride ablation** on FordA, FordB, & ElectricDevices (Sec. 3.4.1, Table 4):

    - At fixed patch size, **increasing stride from heavily overlapping to non‑overlapping windows causes a sharp monotonic drop on FordA/B** & a milder but consistent drop on ElectricDevices.
    - At fixed stride, varying patch size has only a moderate effect.

    This strongly suggests that the main failure mode is coarse temporal downsampling, not PMA itself. Using the small‑stride configuration suggested by this study, we retuned FordA/B & updated Table 1: PMT is now competitive or slightly better than TS2Vec+SoftCLT at 1% labels on both FordA & FordB, in line with expectations for patchified encoders whose labels depend on short‑lived events.

3. **Scope of evaluation (M7TR, iV7V)**

    Our study focuses on **low‑label time‑series classification** on seven benchmarks (HAR, Epilepsy, Wafer, FordA, FordB, POC, ElectricDevices), spanning diverse domains & sequence lengths (80–500) with ≈21.5M training time steps. Within this regime we emphasize **depth of analysis**:
    - Main 1% / 5% linear‑probe results vs strong SSL baselines.
    - Loss‑component ablations (Tables 2–3).
    - Patch/stride sensitivity on FordA/B & ElectricDevices.
    - Architectural ablations including vanilla Transformer, windowed Transformers, xLSTM, & Transformer‑XL under a shared SSL setup (Appendix F, H).
    - Qualitative analyses of token & memory representations (Sec. 3.3, App. J) & a compute analysis (App. E).

    We agree that forecasting, anomaly detection, & transfer learning are important directions. We now **make this explicit** in the Limitations section (Appendix I) & position them as natural follow‑ups rather than requirements for the present contribution.

### Conclusion

The main issue -- **novelty, tokenizer sensitivity on FordA/B, comparisons to recurrent/memory baselines, & evaluation scope -- have all been directly addressed** with targeted experiments & clarifications in the revised manuscript.

Two reviewers (xeyo, iV7V) already view PMT as novel, timely, & technically sound; after the new experiments, **iV7V explicitly recommends acceptance, stating that "the paper has been largely consolidated with the newly provided experiments & polished writing."** M7TR was initially cautious but writes that they "would not mind if the paper is accepted," & their key concerns (novelty, FordA/B behavior, memory baselines) are now empirically addressed.

Because reviewer scores were reverted to their pre-disucission values after the OpenReview incident, the **numeric ratings no longer fully reflect the post‑rebuttal state of the paper.** We understand that the scores are only one signal and respectfully ask the AC to also base their decision on the revised manuscript & the updated reviewer stance, particularly iV7V’s explicit recommendation for acceptance.

---

### Meta-Review · Area_Chair_VEAq · 2025-12-24

**Summary:**

This paper presents the Progressive Memory Transformer (PMT), which is a memory-augmented transformer backbone with a writeable memory bank. Along with a hierarchical contrastive protocol on token, window, and full-sequence levels, PMT achieves promotion in the limited labeled scenario on the classification tasks.
The main concerns of reviewers include the inconsistent empirical superiority, insufficient evaluation and limited novelty:

-	Regarding the model performance, the authors justify the effectiveness of PMT based on ablations of stride size and demonstrate that a small‑stride configuration can lead to better performance.

-	As for the evaluation, the current paper is limited to the classification task. During the rebuttal, the authors clearly discuss this issue in the limitations section, which makes the statement more rigorous. However, I am not convinced by this since the title still indicates a general scope, which is “for Time Series”, not “for time series classification”. It is also widely acknowledged that contrastive learning is good at coarse-scale tasks, like classification, and the mask modeling is better in fine-grained tasks, e.g., forecasting. If the paper only tests one task, I cannot justify whether this is an effective method for time series representation learning.

-	As for the novelty, the authors provide a detailed discussion about the related work. Although PMT is a relatively new architecture (but complicated), I think the hierarchical contrastive protocol is not new (please check [1]), and makes the whole framework combination and incremental. Thus, I think the novelty issue is still not well resolved.

Besides, after reading this paper thoroughly, I think the current title only highlights the architecture, while the final performance is actually guaranteed by the contrastive learning framework. Thus, although the performance seems good, the overall insight is not well presented and supported. As for the review, the two positive reviewers are with lower confidence, and the negative one is more confident.

Considering all the above factors, I cannot recommend acceptance.

[1] Zhao et al., HiMTM: Hierarchical Multi-Scale Masked Time Series Modeling with Self-Distillation for Long-Term Forecasting, CIKM 2024

**Reviewer Concerns:**

As described above, the resolved concerns and their corresponding rebuttal include:

-	The difference between LSTM and memory-augmented Transformer. The authors highlight that the motivation of this work is to extract mid-range states, which can be supervised by contrastive learning. Besides, a performance comparison is also provided.

-	Ablation of the proposed writable memory. The authors made a detailed ablation in Appendix F, where the benefits from writable memory are more significant.

The following concerns still remain:
-	The novelty of the proposed method. As mentioned before, I think the architecture part and the hierarchical contrastive loss seem combinational.

-	The hierarchical PMA appears to be heavy. Note that if your pre-training backbone is too complex, it is really hard for other researchers to use your pre-training model in their own work. I would like to recommend that the authors reconsider the scope of this paper. Is the main contribution about architecture or a self-supervised framework? Similar concern to the novelty, where the whole framework is too complex.

-	Extension to other types of tasks. I think this is a necessary part for a self-supervised learning work on time series.

**Reviewer Scores:**

(1) Reviewer xeyo (initial score: 6, confidence 3). The authors have provided a detailed experiment on the “patchification loss” and added a new discussion about the Sequence Complementor (Chen et al., AAAI 2025). Thus, I think the reviewer will keep his/her original score.

(2) Reviewer hY4L (initial score: 6, confidence 3). The reviewer has actively posted his/her feedback to the rebuttal. However, regarding the reviewer’s concern about “limited to classification” and “heavy PMA”, I do not think the authors gave a sufficient response. As for the PMA design, I think it is too complicated, not about efficiency, but in the design aspect, which can affect its impact on future research.

(3) Reviewer htEG (initial score: 4, confidence 4). The reviewer raised some concerns about the novelty and evaluation. Although the authors add some discussion about the position of this work, I think the design in architecture and contrastive loss seems a combination. Thus, I do not think the reviewer would raise his/her score.

---

> ### Public Comment · ~Tord_Sture_Stangeland1 · 2026-03-06
> **Public comment on factual discrepancies in the meta-review**
>
> We respect the outcome of the review process. However, for the public record, we would like to note several factual discrepancies between the meta-review and the review discussion for this submission.
>
> (i) Reviewer identifiers. The meta-review refers to reviewer identifiers that do not match the identifiers shown in this submission’s public discussion; only one of the three cited identifiers matches the visible review record.
>
> (ii) Reviewer stance after rebuttal. The meta-review states that the rebuttal did not sufficiently address the concerns of the reviewer who “actively posted feedback” on the “limited to classification” and “heavy PMA” points. However, the only post-rebuttal public reviewer comment states that most concerns were adequately addressed and explicitly recommends acceptance. This is not a minor change in emphasis, but the opposite of the reviewer’s final public stance.
>
> (iii) Meaning of “heavy PMA.” The meta-review reframes “heavy PMA” as a design-complexity concern. In the public review discussion, this point was raised in the context of the paper’s original “lightweight” framing and requests for lightweight-baseline comparisons. Our revision responded by explicitly narrowing the claim: Appendix E states that the compute analysis is intended to characterize PMA overhead relative to a matched vanilla Transformer implementation, not to claim superior efficiency over Transformer-XL or PatchTST.
>
> (iv) Novelty citation. The meta-review cites HiMTM to support the claim that our hierarchical contrastive protocol is “not new.” HiMTM is a hierarchical masked-modeling/self-distillation paper for long-term forecasting, not a hierarchical contrastive-learning method, so it does not directly support that specific claim.
>
> Taken together, these discrepancies materially undermine confidence that the meta-review faithfully reflects this submission’s review record.

---

### Decision · Program_Chairs · 2026-01-26

Reject